# Molecular and Kinetic Models for Pore Formation of *Bacillus thuringiensis* Cry Toxin

**DOI:** 10.3390/toxins14070433

**Published:** 2022-06-24

**Authors:** Haruka Endo

**Affiliations:** Department of Integrated Bioscience, Graduate School of Frontier Sciences, The University of Tokyo, Kashiwa 277-8562, Japan; haruka.endo@riken.jp

**Keywords:** *Bacillus thuringiensis*, Cry protein, receptor, pore formation, oligomerization, ABC transporter, 12-cadherin domain protein, binding kinetics

## Abstract

Cry proteins from *Bacillus thuringiensis* (Bt) and other bacteria are pesticidal pore-forming toxins. Since 2010, when the ABC transporter C2 (ABCC2) was identified as a Cry1Ac protein resistant gene, our understanding of the mode of action of Cry protein has progressed substantially. ABCC2 mediates high Cry1A toxicity because of its high activity for helping pore formation. With the discovery of ABCC2, the classical killing model based on pore formation and osmotic lysis became nearly conclusive. Nevertheless, we are still far from a complete understanding of how Cry proteins form pores in the cell membrane through interactions with their host gut membrane proteins, known as receptors. Why does ABCC2 mediate pore formation with high efficiency unlike other Cry1A-binding proteins? Is the “prepore” formation indispensable for pore formation? What is the mechanism underlying the synergism between ABCC2 and the 12-cadherin domain protein? We examine potential mechanisms of pore formation via receptor interactions in this paper by merging findings from prior studies on the Cry mode of action before and after the discovery of ABC transporters as Cry protein receptors. We also attempt to explain Cry toxicity using Cry–receptor binding affinities, which successfully predicts actual Cry toxicity toward cultured cells coexpressing ABC transporters and cadherin.

## 1. Introduction

Cry protein is one of the classification groups of pesticidal proteins from bacterial pathogens [1]. The pesticidal protein kills host epithelial cells, allowing bacteria to penetrate and multiply in the hemocoel. According to the Bacterial Pesticidal Protein Resource Center [2], the Cry protein group has 79 subfamilies and over 700 members. Most Cry proteins have been discovered in *Bacillus thuringiensis* (Bt), a pathogen of insects and nematodes. Other bacteria such as *Paenibacillus popilliae* and *Paraclostridium bifermentans* produce several Cry proteins [3,4]. Each Cry protein kills a distinct range of insect larvae (mostly caterpillars, beetles, and mosquitos) and nematodes (reviewed in [5]). Bt and its Cry genes have been used as biopesticides for roughly 100 years and as a gene source for insect-resistant transgenic crops for 25 years because of their restricted pesticidal spectrum.

Cry proteins are classified as α-pore-forming toxins whose alpha helices form pores on the cell membrane. Cry proprotein (~130 or ~75 *m*/kDa) is produced as a parasporal body during sporulation. In the gut of hosts, proprotein is solubilized and processed into its activated form (~60 *m*/kDa) by gut digestive fluids. The three-dimensional structure of activated Cry protein comprises three highly conserved domains. Domain I, the N-terminus domain, is involved in pore formation and has a colicin-like fold with seven alpha helices. The middle domain, Domain II, has a beta-prism structure and is in charge of receptor interaction and pesticidal specificity. The beta-sandwich structure of Domain III, the C-terminus domain, is critical for receptor interaction and structural stability. Cry genes are encoded on plasmids and appear to have been diversified through domain swapping via homologous recombination to adapt to their host and expand their host range [6]. On the basis of the conserved structure, the Cry protein family’s mode of action must be based on the same underlying principle.

The activated Cry protein binds to multiple membrane proteins and glycolipids on the apical membrane of host gastrointestinal cells. These binding targets include “receptors” that mediate Cry toxicity by binding to the Cry protein. The receptor interactions cause the Cry protein to undergo a dynamic irreversible conformational shift, transforming it from a soluble protein binding to receptors to a membrane-embedded form [7]. The conformational transition includes domain I partially unfolding to expose a hydrophobic helical hairpin, allowing its entry into the cell membrane. Although the three-dimensional structure of the Cry protein pore has yet to be determined, it is thought to be a Cry oligomer centered on the helical hairpins such as an umbrella-like structure postulated in colicin A toxin with structural similarities to the domain I Cry proteins [8]. For example, Cry1A pores are thought to be tetrameric [9]. In the colloid-osmotic lysis model proposed by Knowles and Ellar [10], the Cry pores are permeable to small solutes such as K^+^ and cause an imbalance of osmotic pressure between the outside and inside of the cell, resulting in water influx, cell swelling, and necrotic cell death. This model was further supported by a study showing that the cell death induced by Cry protein is determined by water influx through aquaporins, i.e., water channels [11].

Cry protein’s pesticidal specificity is determined by unique interactions between the Cry protein and receptors [12]. Cry protein receptors have been identified using biochemical screening on the basis of binding properties and genetic mapping of Cry resistance genes. Their mode of action is discussed here using Cry1Aa, Cry1Ab, and Cry1Ac proteins as models because research on the three proteins is the most advanced in the family. Many Cry1A-binding proteins have been found through biochemical screening of the midgut of lepidopteran insects, including aminopeptidase N (APN), 12-cadherin domain protein (hereafter simply referred to as cadherin), and alkaline phosphatase (ALP) (reviewed in [13]). Note that, in principle, the receptor candidates identified by this method include simple Cry-binding proteins that do not play a role in pore formation. In this perspective, “Cry protein receptors” are defined as binding targets of Cry proteins that directly mediate pore formation and distinguishing them from mere Cry-binding proteins.

Cadherin is the only receptor previously identified by binding studies whose role has been proven by genetic studies of many Cry1A-resistant lepidopteran strains. By using linkage mapping, Gahan et al., discovered that over 10,000-fold Cry1Ac resistance in the *Heliothis virescens* YHD2 strain was connected to cadherin [14]. Nevertheless, the cadherin mutation did not prevent irreversible binding and pore formation [15]. Despite their high binding affinity to Cry1A proteins [16,17,18] and oligomerization-inducing activity [19,20,21], cadherins from lepidopteran insects confer low Cry1A susceptibility to cultured cells [22,23,24,25]. Previous research on Cry1A resistance produced by cadherin mutations in other lepidopteran species has repeatedly demonstrated the importance of cadherin [26,27,28]. However, clearly, cadherin alone could not explain the Cry1A mode of action.

Interestingly, the sequential binding model proposed by Bravo et al. [29] appeared to fill major gaps in classical research. In that model, activated Cry1A protein monomers bind to cadherin to oligomerize and subsequently, Cry oligomers, or “prepores” bind to APN, culminating in their insertion into the cell membrane and pore formation [29,30]. Cadherin and APN play the roles of “mediating Cry oligomerization to form prepore” and “mediating membrane insertion of prepore to form pore” in this scenario. Nevertheless, as discussed below, the role of APN in mediating pore formation remained in question.

In 2010, the ABC transporter subfamily C2 (ABCC2) was identified to be a responsible gene for high resistance to Cry1Ac protein in the *H. virescens* YHD3 strain by Gahan et al. [15]. In that same period, an independent positional cloning effort in a Cry1Ab-resistant *Bombyx mori* strain yielded the same target [31]. Since then, increasing literature indicates a central role of ABCC2 in the mode of action of Cry1A and Cry1F proteins in many lepidopteran insects (see [32,33] for detailed reviews). Importantly, in vitro investigations have shown that expression of ABCC2 in cultured cells is sufficient to confer high susceptibility [34] because of its high activity in pore formation [21]. The discovery of ABCC2 prompted us to reconsider the traditional model, which explained the pore formation mechanism using cadherin and APN. High Cry resistance by ABCC2 mutation indicates that cadherin and APN sequential binding, if any, is not a primary mechanism in mediating Cry toxicity. Instead, ABCC2 alone makes a major contribution by mediating Cry oligomerization, membrane insertion, and cell death. Moreover, the discovery of the receptor, which is directly involved in pore formation and whose mutation results in high resistance, provides conclusive evidence for the conventional killing model on the basis of pore formation and osmotic lysis [10]. Interestingly, when ABCC2 and cadherin are coexpressed in the same cell, they exhibit more susceptible to Cry1A proteins in comparison to ABCC2-expressing cells [34,35] based on synergistic cooperation between the two receptors in helping pore formation [21]. Lack of synergistic cooperation seems to explain Cry1A resistance caused by cadherin mutation [21]. However, the molecular mechanism of the synergism that enables pore formation in lower Cry concentration is unknown.

Aside from ABCC2, researchers discovered that other insect ABC transporters could function as Cry protein receptors. ABCC3, a possible paralog of ABCC2 in lepidopteran species, also functions as a Cry1A receptor [36,37,38], but in most cases has a smaller role compared with ABCC2. Notably, depending on the insect species and Cry proteins used, the role of ABCC3 is sometimes equal to or higher than that of ABCC2. For example, double knockout of ABCC2 and ABCC3 caused a high level of Cry1Ac resistance in *Helicoverpa armigera* [39], Cry1Ac and Cry1F resistance in *Plutella xylostella* [40,41], and Cry1F resistance in *B. mori* [42], while single knockout of ABCC2 did not. More surprisingly, ABC transporters from different subfamilies, ABCA2 and ABCB1, were associated with Cry2A and Cry3 resistance in *H. armigera* and *Chrysomela tremula* [43,44]. The fact that phylogenetically distant Cry proteins use various insect ABC transporters as receptors indicates that the Cry protein family has adapted to host ABC transporters.

Therefore, the conventional understanding of Cry protein’s mode of action has progressed over the last decade. Nevertheless, the process of pore formation via receptor interactions is still poorly understood. We compare ABCC2 and cadherin to explore what occurs on the cell membrane from receptor interaction to pore formation and the role of receptors mediating Cry toxicity and present a current molecular model for Cry protein pore formation via receptor interactions. We also aim to explain Cry toxicity in binding kinetics, including synergism between ABCC2 and cadherin.

## 2. Receptor Interactions

### 2.1. Role of Receptors

First, why do Cry proteins require receptors to exhibit toxicities? Activated Cry proteins can form pores on receptor-free artificial lipid bilayers [45]. Nevertheless, receptors in the cell membrane are required to form enough pores efficiently for killing cells and insects. The action of binding to Cry proteins and gathering them from gut fluid to the surface of the target cell membrane is required for efficient interactions to form pores. However, even with high affinity, binding to the Cry protein does not always result in pore formation. The structures of activated Cry protein are expected to have evolved within constraints on maintaining water solubility and protease resistance, resulting in hydrophobic helices in domain I that insert into the cell membrane and form pores located inside the molecule. For efficient pore formation, domain I of Cry protein must be partially unfolded by binding to a receptor near the cell membrane. With this ability causing partial unfolding should distinguish Cry receptors from Cry-binding proteins. ABCC2 and cadherin are considered as Cry1A receptors in this context because they can cause Cry oligomerization, pore formation, and confer cultural cells Cry1A susceptibility. As cadherin has quite lower activity in inducing pore formation than ABCC2, and enhancing ABCC2-mediated toxicity is likely the dominant role of cadherin in the Cry mode of action, it would be more accurate to define cadherin as an “accessory receptor”. The switch for the conformational change is located on domain II of Cry1A proteins [17,46], which serves as the binding site for ABCC2 and cadherin [16,17,18].

### 2.2. Comparison of ABCC2 and Cadherin as a Cry Protein–Receptor

#### 2.2.1. Generality as a Receptor across Cry Protein Family

ABC transporters appear to be a shared receptor for Cry proteins. Aside from Cry1–ABCC2/C3, Cry2–ABCA2, and Cry3–ABCB1, other Cry subfamily and ABC transporter combinations may be discovered. Cry8Ca, generally known as a coleopteran-killing protein, for example, exerts its toxicity through *B. mori* ABCC2, *Spodoptera exigua* ABCC3, and *Tribolium castaneum* ABCC4A [34,37]. This indicates that ABC transporters serve as Cry8Ca receptors in leaf beetles and scarab beetles with high Cry8Ca susceptibility. If all Cry proteins use ABC transporters as their main receptors, Cry protein affinities and specificities toward ABC transporters should explain their pesticidal spectra and reflect their history of host adaptation. Since no obvious orthologs were found in Diptera, Coleoptera, or Hymenoptera so far, ABCC2 and ABCC3 are thought to be lepidopteran-specific ABC transporters [37]. This appears to explain the toxicity of Cry1A and Cry1F proteins to lepidopterans. It is currently unknown whether nonlepidopteran and noncoleopteran insects have orthologs of lepidopteran ABCA2 and coleopteran ABCB1 that function as Cry protein receptors. The cross-order activity of Cry2 proteins in Lepidoptera and Diptera suggests that a dipteran ABC transporter functions as a Cry2 receptor. Nevertheless, why were ABC transporters selected as target proteins of Cry proteins? The answer may be that ABC transporters generally have a high potential for helping Cry protein pore formation.

Conversely, cadherin seems to be an uncommon receptor across the Cry protein family. The AZP-R strain of *Pectinophora gossypiella* with cadherin null mutation [26] is resistant to Cry1Aa, Cry1Ab, and Cry1Ac, but not to Cry1Bb, Cry1Ca, Cry1Da, Cry1Ea, Cry1Ja, Cry2Aa, and Cry9Ca [47]. Cadherin knockout in *Spodoptera frugiperda* and *B. mori* did not affect Cry1F susceptibility [41,48]. CRISPR/Cas9-mediated cadherin knockout in *H. armigera* caused Cry1Ac resistance but no significant change in Cry2Ab susceptibility [49]. Cadherin silencing did not decrease Cry3Aa susceptibility in *Diabrotica virgifera* [50] and Cry3B susceptibility in *Leptinotarsa decemlineata* [51]. Cadherin, sometimes, does not even mediate Cry1A toxicity. Cadherin knockout in *S. frugiperda* did not affect Cry1Ab susceptibility [48]. Cadherin from *Trichoplusia ni* does not bind to Cry1Ac [52]. These examples suggest that cadherin is not a general Cry receptor, but instead, a Cry1A receptor for some lepidopteran species so far. They also suggest that the synergistic cooperation in Cry1A toxicity between ABC transporters and cadherin is quite rare across the mode of action of Cry families. With regard to beetles and mosquitos, testing Cry susceptibility of cadherin knockout larvae is needed to determine the role of cadherin.

#### 2.2.2. Binding Sites on Cry1Aa

Using the data from [53], Figure 1 depicts the relative importance of Cry1Aa residues in binding to BmABCC2 and *B. mori* cadherin (BmCad). Cry1Aa binding to BmABCC2 and cadherin is affected by the deletion of loop 3 (441–444 aa) and a double cysteine mutant R281C-I369C, in which the cavity between domain I and domain II loop 2 is impaired. These areas are close to the boundary between domains I and II, including the Asp222-Arg281 salt bridge, one of the four domain I–II salt bridges. They may serve as the switch triggering partial unfolding of domain I and oligomerization. R366 and R368 are responsible for binding to BmABCC2 and are located near the Arg234-Glu274 salt bridge. Thus, the ABCC2-binding site on Cry1A is close to the two domain I–II salt bridges. Meanwhile, the BmCad-binding site contains domain II loops 1–3, and all three loops contribute equally to binding affinity.

#### 2.2.3. Correlation between Binding Affinity to Cry Protein and Mediating Toxicity

Does the pore formation-helping activity of ABC transporters correlate with binding affinity? Figure 2 depicts a scatter plot of ABC-mediated Cry toxicities and their binding affinities. The Pearson correlation coefficient was calculated using data from seven different combinations of a Cry protein and an ABC transporter acting as its receptor [18,37,38,41], demonstrating the strong correlation (Figure 2). This suggests that ABC-mediated Cry toxicity is dependent on binding affinities. Despite having a nearly identical binding affinity to BmABCC2, the toxicities of Cry1Aa and Cry1Ab are ~100-fold different. This means that even when the same amount of Cry protein binds to ABCC2, the number of pores generated varies, implying that pore formation efficiency varies across Cry–ABC combinations. Pore formation efficiency via ABC transporters (PE_ABC_) is defined as binding affinity/ABC-mediated toxicity (e.g., K_D_/EC_50_). Hence, ABC-mediated Cry toxicity can be calculated as binding affinity/pore formation efficiency.

We further analyzed data from Cry1Aa mutants [18,53]. Those with mutations in the putative ABCC2-binding site exhibit a binding affinity-dependent decrease in BmABCC2-mediated toxicity (Pearson R^2^ = 0.9019, *p* < 0.0001) (Figure 3A). Other Cry1Aa loop mutants clearly exhibit a binding affinity-independent decrease (Figure 3A). These mutants drastically decrease PE_ABCC2_, whereas the ABCC2-binding site mutants do not so much (Figure 3B). Particularly, the loop 2 mutants, D2 (371–376 aa deletion) and ^368^AAA^371^ show a >10,000-fold decrease in PE_ABCC2_ (Figure 3B), suggesting that Cry1A loop 2 may be crucial to the pore formation process. Similarly, a previous study showed that Cry1Ab loop 2 mutants named D2 (370–375 aa deletion), F371A, and G374A exhibit quite lower toxicity toward *M. sexta* [54], likely due to a decrease in PE_ABCC2_. However, the mechanism by which Cry1A domain II loop 2 affects pore formation efficiency is completely unknown. Figure 3C depicts Cry1Aa residues specifically affecting PE_BmABCC2_, and commonly affecting PE_BmABCC2_ and PE_BmCad_.

Given the low and frequently undetectable nature of cadherin-mediated toxicity, there has been a notable lack of data on cadherin-mediated toxicity and binding affinity in the same species or among Cry families. There are examples of Cry1Aa mutants combined with BmCad–TBR (TBR, toxin-binding region). The BmCad-binding affinities of Cry1Aa domain II loop mutants, which included six mutants with ~10–50-fold higher binding affinity than wild-type Cry1Aa, did not correlate with BmCad-mediated toxicities [18,55,56].

#### 2.2.4. Contributions to Cry Susceptibility of Insect Individuals

It is plausible that insect susceptibility to Cry protein is basically based on the sum of contributions of each receptor and synergism between receptors to pore formation, although many other factors (e.g., midgut proteases and cell repairing systems) are involved in determining Cry susceptibility. As ABCC2 exhibits remarkable pore formation-inducing activity for Cry1A proteins, it is expected to be the primary susceptibility determinant in general (Figure 4A). When lacking ABCC2, the second determinant such as ABCC3 contributes to Cry1A susceptibility (Figure 4B). Cadherin itself has quite low activity mediating pore formation but contributes to enhancing ABC-mediated pore formation because of the synergism between ABCC2. Accordingly, Cry1A resistance by a cadherin mutation is likely due to lack of the synergism (Figure 4C). When lacking both ABCC2 and cadherin, a higher resistance is often observed [15,57], likely because all synergisms between cadherin and ABC transporters including ABCC3 are missing (Figure 4D) or because cadherin is the second susceptibility determinant. Thus, in spite of its low pore formation-inducing activity, cadherin significantly contributes to Cry1A susceptibility as well as ABCC2 in some cases. Meanwhile, cadherin mutations do not affect Cry1A and Cry1F susceptibilities in some combinations between the insect species and Cry proteins [48,58], suggesting no synergisms between cadherin and ABC transporters in these combinations.

Altogether, Cry resistance caused by a cadherin mutation indicates the existence of a synergism between cadherin and ABC transporter(s) which is the primary determinant of susceptibility to the Cry protein. In contrast, having no high resistance which is caused by a mutation in an ABC transporter does not necessarily mean that the ABC transporter is not involved in Cry susceptibility. When multiple ABC transporters make the almost same contributions, Cry susceptibility is maintained at almost the same level by remaining receptor(s) even if one of them is lacking.

### 2.3. APN and ALP

#### 2.3.1. Roles of APN and ALP in General

Elucidating the role of APN and ALP has been one of the important issues in studies of Cry’s mode of action. According to our definition of Cry receptors, APN and ALP cannot be considered as a Cry receptor at this stage in general. They bind to Cry1A protein domain III with lower affinity (K_D_ = ~100 nM) in comparison to ABCC2 and cadherin (K_D_ = ~0.1–1 nM) and does not induce Cry1A oligomerization [17,29,59]. Heterologous expressions of lepidopteran APNs in cell lines are not sufficient to mediate Cry1A toxicity in most cases [60]. Unlike ABCC2 and cadherin, no APN and ALP mutations were found to be responsible for field-evolved Cry1A resistance so far.

One example that APN confers Cry susceptibility was reported in *H. armigera*, and its APN1 (HaAPN1) expression in Sf21 cells conferred ~30% cell death after 5-h incubation with 3.75 µg of activated Cry1Ac protein [61]. However, the CRISPR-Cas9 mediated knockout of APN1, APN2, and APN5 in this species did not alter larval susceptibilities for Cry1Ac and Cry2Ab proteins [62]. Taking account of high Cry1Ac resistance caused by mutations in ABCC2 (1400-fold) and cadherin (549-fold) in *H. armigera* [49,63], it should be concluded that the contribution of HaAPN1 to Cry1Ac toxicity is negligible. Likewise, the knockout of APN1 in *S. exigua* and APN1 and APN2 in *Aedes aegypti* did not affect Cry1Ac, Cry1Ca, and Cry1Fa susceptibilities or Cry4Ba and Cry11Aa susceptibilities, respectively [64,65]. *Drosophila melanogaster* larvae expressing *Manduca sexta* APN1 (MsAPN1) were susceptible to Cry1Ac [66], having been considered as compelling evidence that APN1 is a Cry1A receptor. However, the interpretation of the result should be taken with caution because the larval death did not occur in a dose-dependent manner and 100% mortality was suddenly observed with 50 ng/µL Cry1Ac. Meanwhile, *Drosophila* larvae expressing *P. xylostella* ABCC2 exhibited ~100-fold higher Cry1Ac susceptibility than larvae expressing MsAPN1 in a dose-dependent manner [67]. Use of different heterologous expression systems and APN1-knockout larvae in *M. sexta* will help validate the role of MsAPN1.

How about the role of APN as a “secondary receptor” in the sequential binding model? APN has been believed to receive Cry1A oligomers from cadherin and insert them into the cell membrane. If this is the case, APN should be essential for Cry toxicity. Nevertheless, the significant Cry1A resistance generated by ABCC2 mutations in many lepidopteran insects [15,31,63,68,69] indicates that APN has a much smaller function than ABCC2. Furthermore, given conventional research, results before the discovery of ABCC2 must be reinterpreted. For example, Cry1Ab domain II loop 3 was reported to be important for the sequential binding and affect toxicity [30], but the loop region in Cry1Aa is now also known as the binding site to ABCC2 [53]. Overall, although APN is indeed a binding target of Cry proteins such as Cry1A and its abundant expression in the midgut implies that APN somehow influences the Cry mode of action, the direct contribution of APN in mediating pore formation may be considered negligible.

#### 2.3.2. Role of APNs in *Plutella*

Molecular understanding of Cry1Ac susceptibility in *P. xylostella* is confusing. First, different results of CRISPR-Cas9-mediated PxABCC2 and PxABCC3 knockouts were reported. Liu et al. and Zhao et al. reported that single knockouts of either PxABCC2 and PxABCC3 caused no or several-fold Cry1Ac resistance but the double knockout of them caused >8000-fold resistance [40,41], suggesting that PxABCC2 and PxABCC3 make the almost same contributions in Cry1Ac susceptibility (Figure 5). Functional redundancy of the two ABC transporters in Cry1Ac and Cry1F susceptibility were reported in other lepidopteran species [39,42]. Thus, the results from Liu et al. and Zhao et al. can be reasonably interpreted based on known mechanisms.

In contrast, the results from Guo et al. suggest that different mechanisms based on putative unknown “synergisms” underlie the primary Cry1Ac susceptibility determinant in the same species. They reported that single knockouts of PxABCC2 and PxABCC3 independently caused 413- and 724-fold Cry1Ac resistance [70], suggesting that the primary susceptibility determinant is based on the mechanism requiring both PxABCC2 and PxABCC3 (i.e., the synergism between the two) (Figure 5). However, as discussed later in 3.5.2, a synergism between ABCC2 and ABCC3 is generally considered unlikely. Furthermore, the same group reported that *P. xylostella* APN1 and APN3a were functional receptors for Cry1Ac [71]. Heterologous expressions of the two APNs conferred very low Cry1Ac susceptibility to Sf9 cells (~10% cell death after 24h incubation with 1 µM Cry1Ac) and their CRISPR-Cas9-mediated knockouts independently caused 463- and 346-fold Cry1Ac resistance [71]. This suggests that the two APNs contribute to the primary susceptibility determinant and that the roles of the APNs do not compensate each other, but are rather synergistic. Because the low susceptibility-conferring activities of the two APNs indicates their low pore formation-helping activities, their contributions to the primary susceptibility determinant are expected to be based on synergisms with ABCC2 and ABCC3, as well as the synergism between ABCC2 and cadherin in other species. Recently, the same group reported the quadruple knockout of PxABCC2, PxABCC3, PxAPN1, and PxAPN3a showed >34,000-fold Cry1Ac resistance [72], suggesting that the quadruple knockout disrupts further other synergism(s) requiring other molecule(s) than PxABCC2, PxABCC3, PxAPN1, and PxAPN3a, as the four molecules are suggested to participate in the primary susceptibility determinant. Taken together, a series of these studies suggests that an obligate synergism of PxABCC2, PxABCC3, PxAPN1, and PxAPN3a governs the Cry1Ac susceptibility in *P. xylostella*. However, the roles of the two APNs cannot be determined until the mechanisms underlying many putative “synergisms” are unraveled. 

## 3. Oligomerization, Membrane Insertion, and Pore Formation

### 3.1. Mechanisms of Oligomerization

How does Cry oligomerization occur? Before oligomerization, each monomer must bind to a receptor to expose the helices essential for oligomerization and pore formation, whether in solution or in the cell membrane. Since the Cry1A–cadherin interaction has a 1:1 stoichiometry [73] and the Cry1A–ABCC2 interaction likely has a 1:1 stoichiometry [38], tetramer formation could not occur on a single receptor molecule but could occur in theory on an assembly of Cry1A–receptor complexes and/or receptor-free conformation-changed monomers (Figure 6). Cry1A–receptor complexes are expected to be dominant because of high affinity interactions with ABCC2 and cadherin [16,17,18]. Consequently, we discuss oligomerization and pore formation from the standpoint that Cry–receptor complexes primarily consist of prepore (oligomer) and pore. Note that it is uncertain whether a Cry monomer remains associated with or dissociates from the receptor even after membrane insertion. The possibility that an assembly of receptor-free conformation-changed monomers can form pores in the cell membrane remains.

How does a Cry–receptor complex assemble? Interactions of domain I helices between conformationally changed monomers induce oligomerization. The probability and efficiency of oligomerization are expected to depend on the density of receptor molecule(s) and diffusion speed on the basis of membrane fluidity.

### 3.2. The “Prepore” and Pore

It has been debated whether oligomerization or membrane insertion occurs first in the Cry mode of action. However, because these two actions occur concomitantly in most α-pore-forming toxins [74], this argument may be unnecessary. In fact, both Cry1A monomers and prepore tetramers (generated in most cases by incubation with cadherin fragments) can insert into the cell membrane and form pores [9,75]. Furthermore, there is little evidence that additional structural alterations of prepores are required to form a pore in the cell membrane. For these reasons, it is important to consider the existence ratio of monomers and prepores and their contributions to pore formation.

### 3.3. Pore Formation via ABCC2 and Cadherin

We, likely, do not need to consider the role of prepores in the case of ABCC2-mediated pore formation. It is fair to assume that the domain I helices of a conformation-changed monomer binding to ABCC2 are inserted into the cell membrane. The completion of the assembly of four Cry–ABCC2 complexes directly signifies pore formation (Figure 7). Conversely, prepores are most likely prevalent in cadherin-mediated pore formation. Cry1A incubation with cadherin-expressing cells resulted in oligomer formation but only a trace quantity of pores was observed in contrast to ABCC2-expressing cells [21], indicating that most oligomers hang downward from cadherin without interacting with the cell membrane are prepore. In this case, the probability of prepore insertion is low, likely due to structural limitations caused by binding to cadherin via domain II (Figure 7).

### 3.4. Factors Generating the Difference in Pore Formation Efficiency between ABCC2 and Cadherin

Why does ABCC2 efficiently mediate pore formation of Cry1A proteins? The apparent difference in molecular characteristics between ABC transporters such as ABCC2 and other Cry-binding proteins is the transport activity of substrates. This activity derives energy from ATP hydrolysis and the accompanying conformational change between open and close states, which might underlie the pore formation mechanism via ABCC2. Heckel hypothesized that inserting a Cry prepore into the outward-facing cavity of open state ABCC2 aids in the pore formation in the lipid bilayer [76]. Moreover, it could be hypothesized that a Cry monomer attaches to ABCC2 in the closed state and that the conformational transition of ABCC2 to the open state promotes membrane insertion.

These hypotheses were appealing, but they seem implausible for several of the reasons listed below. First, the introduction of Cry protein helices into the ABCC2 cavity appears physically impossible. Cry1A protein possesses complex cavities in domain II that bind to ABCC2 [53]. The Cry1A binding site in ABCC2 is likely to include the extracellular loop (ECL) 1 and 4 in the closed state [77]. Domain I helices cannot insert into the ABCC2 cavity because the cavity’s entrance is closed and occupied by domain II. Although the hypothesis alludes to the possibility of interaction between ABCC2 and oligomers (prepore), the possible presence of free oligomers in a solution that does not bind to any receptors should be given more consideration, as discussed below. Second, ABCC2 in the close-state fully functions as a Cry1A receptor. BmABCC2 mutants lacking transport activity preserve the receptor activity for Cry1Aa [77]. ABCC2 from a Xentari^TM^-resistant *S. exigua* strain, which lacks a part of the nucleotide-binding domain (NBD) II in the C-terminal, serves as a functional receptor for Cry1A proteins [78]. These mutants have alanine substitutions in the Walker A motif or a deletion in an NBD, and they appear to lack ATP binding, resulting in no dimerization of NBDs and the closed state. These results indicate that the open state is not required for pore formation. Consequently, ABCC2′s transport activity and accompanying conformational change are not involved in its high pore formation-helping activity.

What, then, are the underlying factors? One possibility is the distance from the cell membrane. Since the extracellular region of ABCC2 only comprises a very small part of the protein, the Cry1A–ABCC2 interaction occurs very close to the cell membrane (Figure 8), which could increase the likelihood of membrane insertion. Cry1A–cadherin and Cry1A–APN interactions seem to occur at a greater distance (Figure 8). Another option is receptor localization in lipid rafts, a membrane microdomain enriched in sphingolipid and cholesterol. In fact, monomeric and oligomeric Cry1Ab protein is mainly detected from a detergent-resistant fraction of *M. sexta* brush border membrane vesicles (BBMV) containing lipid rafts [29]. Some studies suggest the importance of lipid raft integrity in the pore formation activity of Cry1Ab toward *H. virescens* and *M. sexta* BBMV [79] and the binding and toxicity of Cry1C to Sf9 cells [80]. Although ABCC2 has not yet been tested for localization, ABC transporters are generally found in lipid rafts [81]. Therefore, receptor localization in lipid rafts may improve Cry oligomerization and membrane insertion efficiency. However, it is unknown whether the importance of lipid rafts stems from the lipid preference of Cry protein on pore formation or from the proper conformation of ABC transporters in the cell membrane to serve as a Cry receptor.

Why does cadherin mediate Cry1A pore formation so ineffectively? Despite its high binding affinity for Cry1A proteins and oligomerization-inducing activity, both of which appear to correspond to ABCC2, cadherin’s pore formation-helping activity was ~5000-fold lower than ABCC2 in *B. mori* [21]. Cry1Aa pore formation efficiency (K_D_/EC_50_) is more than 90-fold lower via BmCad than via BmABCC2. In terms of distance from the cell membrane, the Cry–cadherin interaction occurs relatively far from the cell membrane when compared with the Cry–ABCC2 interaction because of the length of cadherin repeat(s) and membrane–proximal extracellular domain (MPED) (Figure 8). Cadherin is not normally localized in lipid rafts during membrane localization [29].

### 3.5. Synergism between ABCC2/C3 and Cadherin in Inducing Efficient Pore Formation

#### 3.5.1. A Molecular Model of the Synergism

The molecular mechanism of ABCC2 and cadherin synergism in assisting Cry1A pore formation remains unknown. The essence of the synergism is that ABCC2 can mediate pore formation using abundant Cry1A proteins binding to cadherin. Some researchers believe that cadherin’s sequential binding partner is ABCC2 rather than APN, and that ABCC2 is responsible for receiving tetrameric prepores from cadherin and inserting them into the cell membrane [35,76,88]. Nevertheless, there is no convincing evidence to support this assertion thus far. First, does the sequential binding of ABCC2 and cadherin occur? It should happen only rarely. When a tetrameric prepore is formed, four Cry1A–cadherin complexes assemble, resulting in tetravalent binding to cadherin. Because of the accumulated strength of monovalent interactions or avidity, such polyvalent interactions exhibit an apparent extremely high affinity. The binding affinity of the Cry1Ab oligomer to APN, for example, is ~200-fold greater than that of the monomer [29]. Consequently, tetramers are expected to have a much higher affinity than monomers whose K_D_ for Cry1A is ~0.5 nM, and hardly ever dissociate from cadherin. Furthermore, because their Cry1Aa-binding sites overlap [18], cadherin and ABCC2 compete for Cry1Aa binding. Thus, the likelihood of transferring an entire oligomer from cadherin to ABCC2 is quite low.

The synergism could be explained by the interactions between Cry1A–ABCC2 and Cry1A–cadherin complexes via domain I helices rather than sequential binding via domain II. When four Cry–cadherin complexes form a tetramer, ABCC2 finds it difficult to insert it into the cell membrane. However, it is possible in the case of a single Cry1A–cadherin complex (monomer) and assemblies of two to three Cry1A–cadherin complexes (dimer and trimer). Interactions between partial-unfolded domain I of Cry receptor complexes, which is the same force that causes Cry oligomerization, may draw domain I helices of cadherin-binding Cry1A protein into the cell membrane, resulting in pores comprising four Cry1A–ABCC2/Cry1A–cadherin complexes (Figure 9). The ratio of Cry1A–ABCC2: Cry1A–cadherin that can form the Cry pore could theoretically be 4:0, 3:1, 2:2, or 1:3. 

#### 3.5.2. Explanation and Prediction of Synergism-Mediated Cry Toxicity Based on Binding Kinetics

The magnitude of the synergistic effect varies across ABC-cadherin combinations and across Cry proteins. For instance, the BmABCC2-BmCad synergism causes 10-, 1000-, and 100-fold higher toxicity of Cry1Aa, Cry1Ab, and Cry1Ac when compared to BmABCC2 alone, and no synergism was observed in Cry1Fa [34,42]. What is the kinetic principle underlying highly efficient pore formation by the synergism? Do binding affinities of Cry protein to ABCC2 and cadherin explain the synergism-mediated toxicity? Figure 10A depicts a schematic model of the relationship between Cry concentration and Cry binding to cells expressing ABC, cadherin, and ABC/cadherin. When the Cry concentration is K_D ABC_ and K_D cad_, 50% of the receptors are occupied by the Cry protein. Thus, the amount of binding Cry protein at the EC_50_ toward ABC-expressing cells (EC_50 ABC_) is required for killing 50% of cells via ABC transporters.

On the assumption that ABC-binding proteins can form pores with cadherin-binding proteins at the same efficiency as among ABC-binding proteins, the same binding amount of Cry protein kills 50% of ABC/cadherin coexpressing cells. When K_D cad_ is comparable with or lower than K_D ABC_, the binding amount of Cry protein required for killing 50% cells is achieved at a lower concentration (EC_50 ABC/cad_) when compared with EC_50 ABC_ (Figure 10A). This may be the basic principle of the synergism between ABC transporters and cadherin, and is thought to be a reasonable explanation for actual examples of the synergism (Figure 10B–D). In the combination of Cry1Aa and BmABCC2, K_D ABC_ and K_D cad_ are comparable, and their synergism could realize the ~10-fold high toxicity (Figure 10B) as explained in Figure 10A. In the combination of Cry1Fa and BmABCC2, K_D cad_ is more than 10-fold higher than K_D ABC_. An increase in the binding amount of Cry protein is not expected at a lower concentration than the susceptibility threshold Cry1Fa concentration of ABCC2-expressing cells, resulting in no synergism (Figure 10C). In the combination of Cry1Aa and BmABCC3, K_D cad_ is low enough to generate the synergism, and actually, BmABCC3 and cadherin exhibit the synergism in mediating Cry1Aa toxicity (Figure 10D). The synergism could occur at lower concentrations in terms of the binding amount of Cry protein. However, relatively high K_D ABCC3_ may limit the potential because there should be a threshold of the Cry amount binding to ABC required for exerting the synergism.

Overall, as a necessary condition for generating the synergism, K_D cad_ must be comparable with or lower than K_D ABC_. As the upper limit of toxicity enhancement by the synergism is determined by K_D ABC_ or K_D cad_, the lower K_D ABC_ and K_D cad_, the higher the synergism-mediated toxicity. Thus, the synergism-mediated toxicity can be represented as K_D ABC_ * K_D Cad_/PE_ABC_ (=EC_50 ABC_ * K_D Cad_). Toxicities predicted using this formula are strongly correlated with the actual synergism-mediated toxicity observed in in vitro assays (R^2^ = 0.9039, *p* < 0.001) (Figure 11), indicating that the Cry–receptor binding affinity explains the synergism-mediated toxicity.

In this context, a synergism between ABC transporters is normally unlikely, as Cry binding affinities correlates with ABC-mediate toxicity (Figure 2), i.e., pore formation-helping activity. The toxicity mediated by the two ABCs should be basically represented by their sum. There can be a synergism between two ABC transporters if they satisfy the following conditions: one ABC transporter exhibits a higher binding affinity than another, but lower pore formation-helping activity such as cadherin, and another ABC transporter with high pore formation-helping activity.

## 4. Model for Pore Formation of Cry Protein via Receptor Interaction

Basically, the killing mechanism of Cry protein appears to be quite simple: the more pores that are generated, the greater the toxicity. We present a model of Cry1A pore formation via receptor interactions. By inducing a conformational change in the Cry protein, ABCC2 mediates membrane insertion of Cry domain I helices. Cry–ABCC2 complexes diffuse on the basis of membrane fluidity, and collisions between Cry–ABCC2 complexes generate dimers, trimers, and tetramers via intradomain I helices interactions, according to the kinetic model for the stepwise Cry oligomerization process [9]. The Cry pore is formed by the tetrameric Cry–ABCC2 architecture. As cadherin mediates the conformational change but the probability of insertion of Cry domain I helices is low, most tetrameric Cry–cadherin complexes contain only a prepore. In the presence of ABCC2, however, domain I helices of a Cry–cadherin complex can insert into the cell membrane by coupling with a Cry–ABCC2 complex, resulting in the synergism in highly efficient pore formation at lower Cry concentration.

The Cry toxicity can be explained by binding affinity to its receptor. The ABCC2-mediated toxicity is represented as K_D ABC_/PE_ABC_. The synergism (ABC/Cad)-mediated toxicity strongly correlates with K_D ABC_ * K_D Cad_/PE_ABC_ (= EC_50 ABC_ * K_D Cad_). In the future, in vivo toxicity toward pest individuals may be predicted if a much more accurate mathematical model is built using our formulas as prototypes by incorporating contributions to other receptors (if exist), the expression levels of receptors, and other factors filling gaps between in vivo and in vitro.

## 5. Future Perspectives

Pore formation mechanisms are a major gap in our understanding of Cry protein mode of action. Three-dimensional structures of Cry pores and Cry receptor complexes are required for a complete understanding of the process, from receptor interactions to pore formation, and to test the hypothesized model we propose here. It is also critical to understand how Cry1A domain II loops contribute to pore formation.

Based on solid mechanistic and kinetic logic, we will be able to design Cry protein variants active against Cry-resistant strains and naturally nonsusceptible pests. There are two examples of successful Cry protein engineering that overcame Cry resistance. First, Cry1Mod proteins, which lack domain I alpha-1 and spontaneously form oligomers, are effective against a wide range of Cry1A-resistant lepidopteran strains, including ABCC2- and cadherin-deficient strains [90,91]. Overcoming ABCC2 resistance does not seem to require prepore formation in the absence of cadherin. Dissecting the molecular mechanism by which Cry1Mod proteins overcome resistance may improve Cry toxicity independent of receptor interactions. It should also be investigated why Cry1Mod is less active in wild-type strains in general [91].

The other example is phage-assisted continuous evolution (PACE), which allowed Cry1Ac protein affinity maturation to *T. ni* cadherin and overcoming resistance [52]. PACE is an epoch-making system for molecular evolution engineering, but few combinations of Cry1A and lepidopteran cadherin have no or low binding affinity. Instead, this method may be useful for Cry proteins that do not use cadherin (e.g., Cry1Fa, Cry2A, and Cry3) to confer the synergism between ABC transporters and cadherin, with the expectation of improved toxicities unless mutants obtained do not lose binding affinity to ABC transporters. Meanwhile, affinity maturation of Cry proteins toward ABC transporters is required to improve Cry toxicity via ABCC2, ABCC3, ABCA2, and ABCB1 and generate novel Cry receptors. All ABC transporters expressed in the host gut cell membrane are potential target proteins. Unfortunately, PACE does not seem applicable for affinity maturation to ABC transporters because the system applies the *Escherichia coli* expression system that is not suitable for the expression of 12-transmembrane helix transporters. An alternative method for Cry engineering toward ABC transporters is required for freely designing Cry toxicity and its specificity.

## Figures and Tables

**Figure 1 toxins-14-00433-f001:**
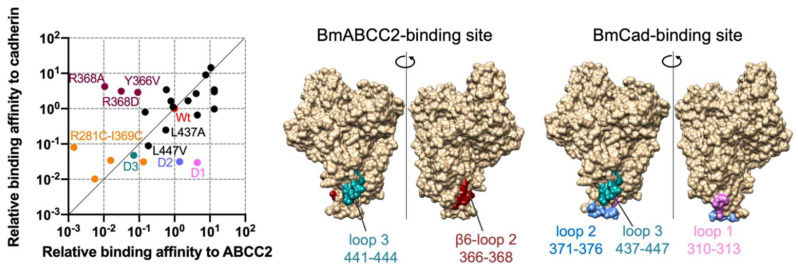
Comparison of Cry1Aa residues responsible for binding affinity to BmABCC2 and BmCad. Relative binding affinity was calculated by K_D wt_/K_D mut_ (the relative value of wild-type as 1). High and low values indicate high and low binding affinity to BmABCC2 and BmCad. Data plotted are derived from [53]. The structures were visualized by UCSF Chimera.

**Figure 2 toxins-14-00433-f002:**
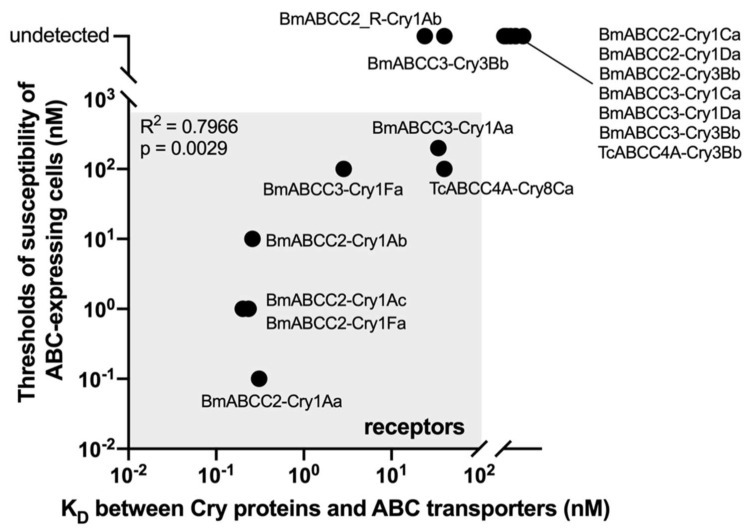
Strong correlation between ABC-mediated Cry toxicities and Cry–ABC-binding affinities. The Pearson correlation coefficient was calculated using Prism 8 (GraphPad) and seven Cry–ABC combinations and an ABC transporter as a Cry receptor. Data plotted are derived from [18,37,38,42].

**Figure 3 toxins-14-00433-f003:**
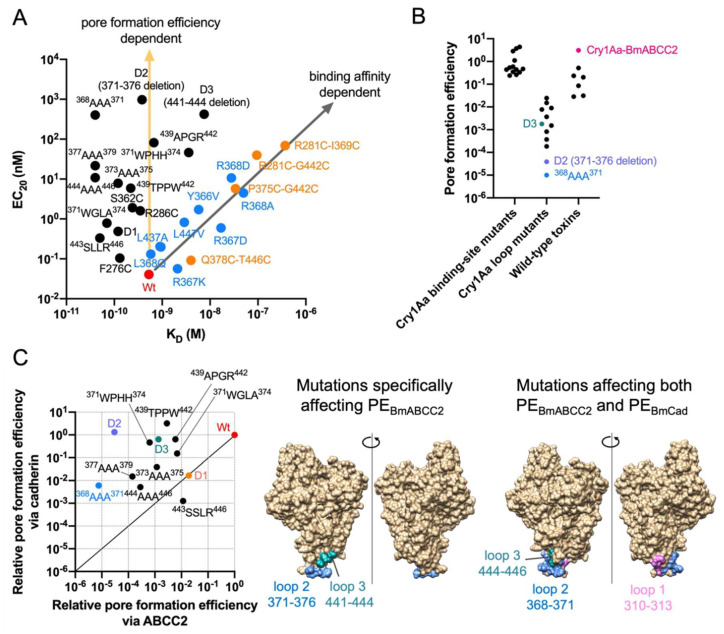
Cry1Aa mutant binding affinity-dependent and pore formation efficiency-dependent BmABCC2-mediated toxicities. (**A**) A scatter plot of BmABCC2-mediated toxicities Cry1Aa mutants (EC_20_) and binding affinities between BmABCC2 and the mutants. (**B**) Pore formation efficiency of wild-type Cry proteins and Cry1A mutants. The pore formation efficiency was calculated by EC_20_/K_D_ for combinations between Cry1Aa/its mutants and BmABCC2, and the Cry susceptibility threshold of BmABCC2 or BmABCC3-expressing cells/K_D_ between other wild-type Cry proteins and BmABCC2/BmABCC3 shown in Figure 1. Data plotted are derived from [18,37,38,42]. (**C**) Comparison of Cry1Aa residues responsible for pore formation efficiency (PE) via BmABCC2 and BmCad. Relative PE was calculated by PE_mut_/PE_wt_ (the relative value of wild-type as 1). Data plotted are derived from [53]. The structures were visualized by UCSF Chimera.

**Figure 4 toxins-14-00433-f004:**
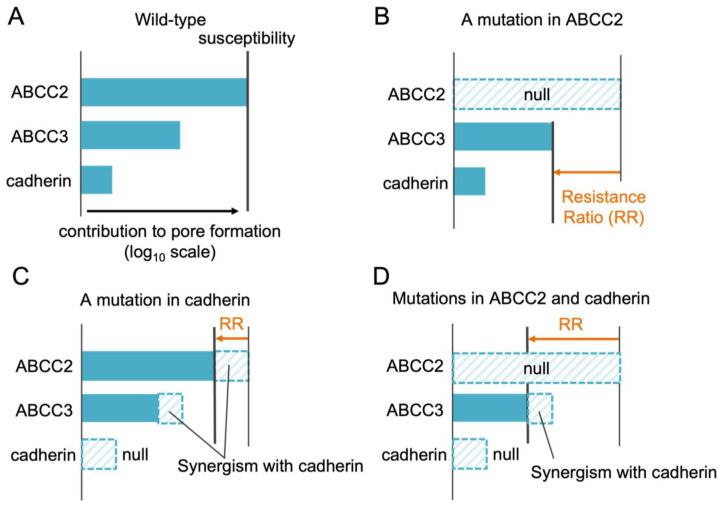
Schematic diagrams of possible mechanisms of Cry1A resistance caused by receptor mutations. (**A**) Wild-type. (**B**) A mutation in ABCC2. (**C**) A mutation in cadherin. (**D**) Mutations in ABCC2 and cadherin.

**Figure 5 toxins-14-00433-f005:**
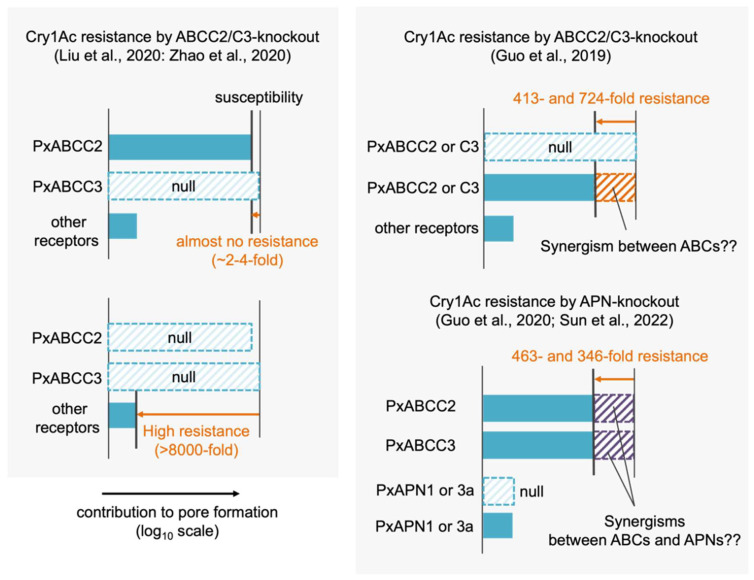
Different contributions of ABCC2, ABCC3, APN1, and APN3a to Cry1Ac susceptibility in *P. xylostella*, suggested by [40,41] (**left**) and [70,71,72] (**right**).

**Figure 6 toxins-14-00433-f006:**
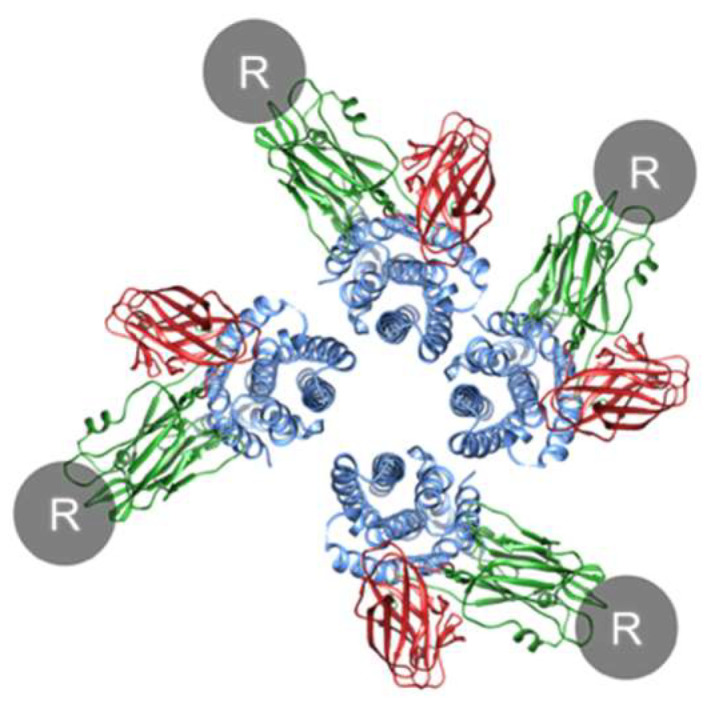
Schematic model of oligomerization. R indicates a receptor. Note that the conformation change is not considered in this figure, and the structure consisting of the Cry pore is not based on experimental evidence but includes an artistic rendering by the author.

**Figure 7 toxins-14-00433-f007:**
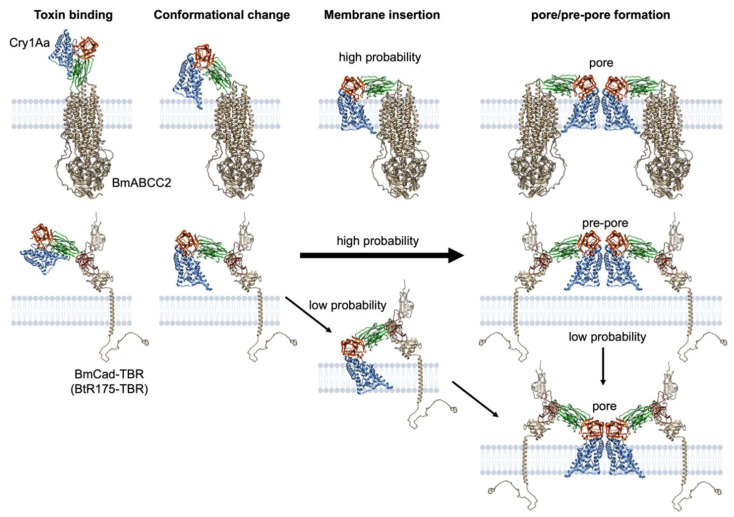
Schematic models for Cry1A pore formation via receptor interaction. Note that the structures of the conformationally changed Cry protein and Cry–receptor complex are not based on experimental evidence but include an artistic rendering by the author. For convenience, pore and prepore formation are explained by two Cry–receptor complexes instead of the four complexes.

**Figure 8 toxins-14-00433-f008:**
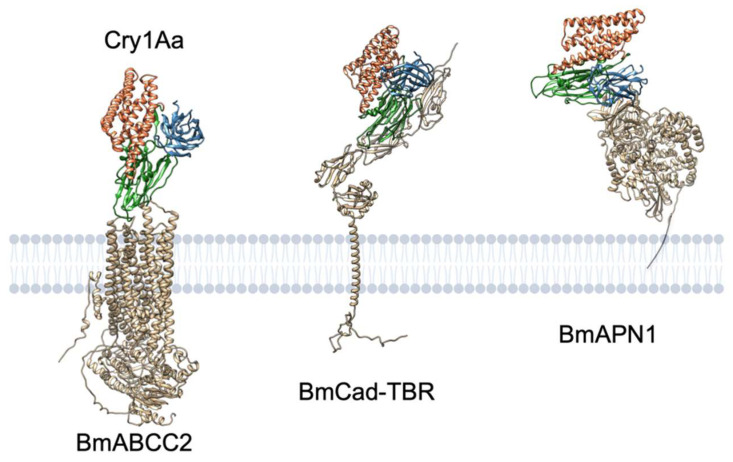
Predicted interactions between Cry1Aa and ABCC2, cadherin, or APN1 from *B. mori*. The three-dimensional structures of BmABCC2, BmCad–TBR (TBR, toxin-binding region), and BmAPN1 were predicted by AlphaFold2 [82] on Google Colaboratory (AlphaFold Colab) [83]. Cry1Aa binding to these proteins was predicted via HADDOCK [84,85] on the webserver (https://wenmr.science.uu.nl/haddock2.4/ (accessed on 3 May 2022)) using putative interaction sites reported by previous studies [18,53,77,86,87]. Note that the shown structures are the first ranked models among the predicted models, but the first rank does not necessarily guarantee precision.

**Figure 9 toxins-14-00433-f009:**
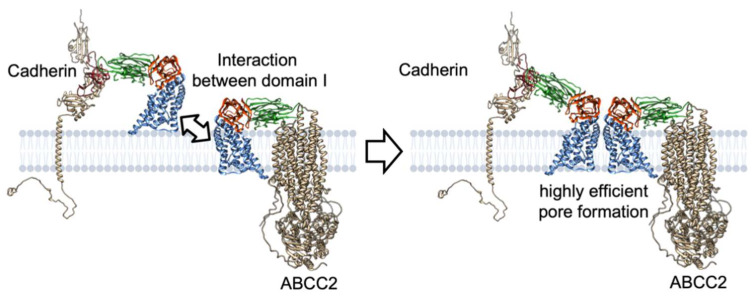
A hypothesized mechanism of the synergism between ABCC2 and cadherin in Cry1A mode of action. Note that the structures of conformation-changed Cry protein and Cry–receptor complex are not based on experimental evidence but include an artistic rendering by the author. For convenience, pore and prepore formation are explained by two Cry–receptor complexes instead of the four complexes.

**Figure 10 toxins-14-00433-f010:**
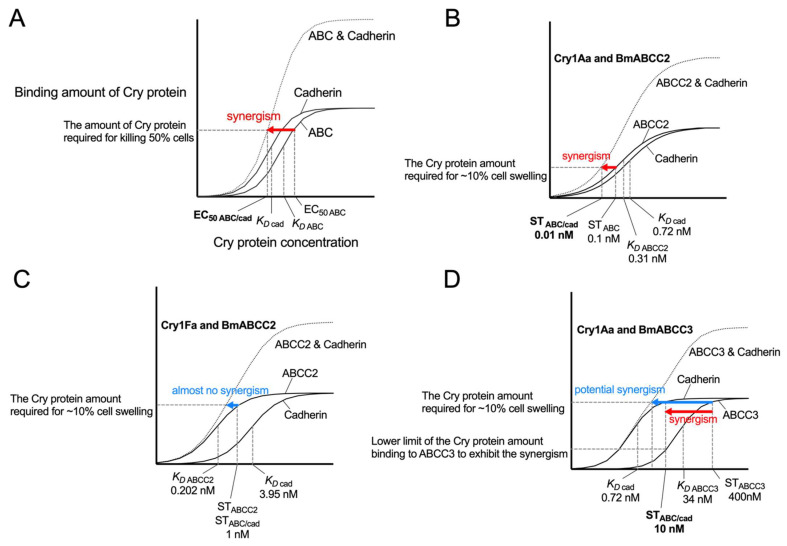
Schematic model for explaining the synergism based on binding kinetics. The binding amount of Cry protein to cells expressing ABC, cadherin, and ABC/cadherin is estimated using sigmoidal curves based on receptor occupancy rate (50% occupancy at K_D_). For convenience, expression levels and Cry-binding properties of ABCC2 and cadherin are assumed to be the same. (**A**) A schematic model. (**B**–**D**) Schematic models explain actual examples of Cry1 toxicity toward Sf9 cells expressing BmABCC2/C3 and BmCad. ST indicates the susceptibility threshold of Cry concentration in which ~10% of cells exhibit swelling. Data used is derived from [18,34,38,42,89].

**Figure 11 toxins-14-00433-f011:**
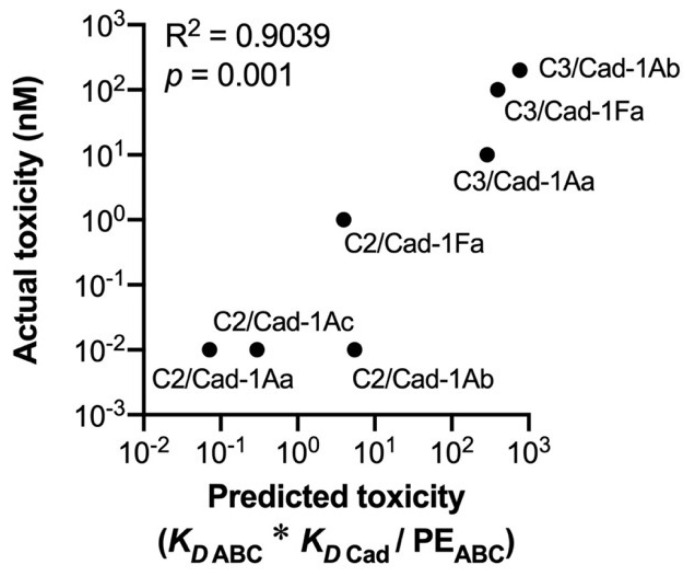
Strong correlation between actual and predicted Cry toxicities toward ABC/Cad coexpressing cells. Actual toxicity is the susceptibility threshold of Cry concentration in which ~10% of cells coexpressing BmABCC2/C3 and BmCad exhibit swelling [18,34,38,42,89]. Predicted toxicity only indicates an index of the toxicity level and does not predict the actual cell susceptibility threshold of Cry concentration (the unit of predicted toxicity is not M but M^2^). The Pearson correlation coefficient was calculated using Prism 8 (GraphPad).

## Data Availability

There is no original experimental data obtained in this study. Numerical raw data used for preparing the figures in this study are available upon request.

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
