# Peer review of "Molecular and Kinetic Models for Pore Formation of Bacillus thuringiensis Cry Toxin"

_toxins, 2022, doi:10.3390/toxins14070433_

Round 1
Reviewer 1 Report
This perspective synthesizes recent advances in mode of action of Bacillus thuringiensis Cry1 toxins and also proposes thoughtful molecular and kinetic models for pore formation by Bt Cry1 toxins. This review is well written and easy to follow. The existing literature is well cited. I strongly agree with your concerns on too many putative receptors in diamondback moth proposed in several recent studies (section 2.3.2). Below I have some comments and suggestions, which hopefully can help improve the review.
- Page 3 lines 120-121. Functional redundancy of ABCC2 and ABCC3 in Cry1Ac mode of action was first reported in Helicoverpa armigera by Wang et al. (PLoS pathogens 2020, 16, e1008427). This literature should be cited.
- Schematic diagrams proposed in Fig. 4 and Fig. 5 are generally correct on different contributions of receptor mutations. Please check and improve the panel C of Fig. 4 (is there synergism between Cad and C3) and the left top panel of Fig. 5.
- Please check and make sure all insect latin names are in italics throughout the review.
Author Response
1. Page 3 lines 120-121. Functional redundancy of ABCC2 and ABCC3 in Cry1Ac mode of action was first reported in Helicoverpa armigera by Wang et al. (PLoS pathogens 2020, 16, e1008427). This literature should be cited.
>Thank you very much for pointing out that we did not cite the important literature. We added Wang et al. as reference 38 (line 126).
2. Schematic diagrams proposed in Fig. 4 and Fig. 5 are generally correct on different contributions of receptor mutations. Please check and improve the panel C of Fig. 4 (is there synergism between Cad and C3) and the left top panel of Fig. 5.
>We appreciate that the reviewer had a look at these figures very closely. We added the ABCC3-cadherin synergism in Fig. 4C and revised the left bottom panel of Fig. 5 to swap PxABCC2 and PxABCC3. According to resistance ratios from Liu et al. and Zhao et al., PxABCC3 has slightly higher contribution than PxABCC2.
3. Please check and make sure all insect latin names are in italics throughout the review.
>Thank you for letting us know. We changed all scientific names to be written in italics.
Reviewer 2 Report
This review summarizes knowledge on the binding and toxicity (as a proxy to pore formation) of Cry1A proteins in Lepidoptera. The authors also provide a hypothetical model combining published data supporting that binding kinetics directly predict pore formation and receptor activity, and that the kinetics of distinct receptors explain cases of synergism. While the idea of binding kinetics being directly related to pore formation is well established in the field, the hypothesis of synergism being dependent on binding kinetics of receptors involved seems novel and supported by the data presented. Since the review and proposed model is built on data with Cry1 proteins and Bombyx mori, I think the title should be revised to be more specific. As written, the current tile is too general and is misleading as the review does not go into detail into other Cry proteins. Suggested title: “Molecular and kinetic model for Cry1 protein pore formation in silkworm (Bombyx mori)”. This reviewer has other relevant points and minor corrections for the authors to consider.
One of the main points is that alkaline phosphatases are recognized as receptors in the literature akin to APNs, but are not considered or mentioned in the review. I recognize there is not much data available, but I consider it deserves to be mentioned as published reports support it is a relevant binding site for toxicity of Cry proteins.
I also have concerns with the model proposed to explain cadherin/ABCC2 synergism and its biological relevance, especially since authors boldly propose that the formulas in 573-577 can be used to predict insect susceptibility. The biggest concern is that the proposed model is entirely built on data from in vitro experiments with cultured cells expressing B. mori receptors. This is an artificial system that may be used to explore some questions, but is probbaly not amenable to explain toxicity in vivo. There are a number of other proteins in the gut known to interact with Cry proteins that would affect the outcomes of the interactions modeled here, and these proteins are not considered in the model (ALP, APN, lipophorin…). For instance, the Cry oligomer bound to cadherin displays high affinity binding to APN and ALP. This would limit the transfer of oligomer from cadherin to ABCC2 proposed in Fig. 9. Overlapping Cry binding sites in cadherin and ABCC2 also present an issue for the model. If binding to ABCC2 is more conducive to pore formation (toxicity), then less productive Cry binding to cadherin would reduce the number of pores formed compared to when ABCC2 is the only receptor (again, this is a limitation of a model based on invitro tests with one or two receptors present). Synergism in this system of two receptors would only be observed when binding sites are saturated or when the affinity of the less favorable receptor (cadherin) is lower (higher Kd) than the more effective receptor (ABCC2). This is in contrast to what authors claim in line 530. In any case, for this system to work the Cry protein must be released from ABCC2 so the receptor is recycled. More importantly, other gut proteins surely would affect this process and are not considered in the model. Authors should consider these limitations and include them in the text.
While the paper is well organized, it is difficult to follow at times due to the style of writing. There are also some typos that need to be corrected for clarity. An important correction is that species names should be always italicized (none of the names are italicized in the text of the version I reviewed).
The authors use “toxin” sometimes referring to Cry1A toxins or Cry toxins, so unless they refer to a specific protein, they should use the plural (Ex: line 36 “Cry toxin is classified…” should be “Cry toxins are classified…”. Also, I would suggest using “insecticidal protein” or “pesticidal protein” instead of “toxin”.
Line 32 “moths” are not larvae (ok to state beetle and mosquito larvae). Change to “caterpillars”.
Line 33 “Cry genes” should be “Cry proteins”.
Line 34 Correct “gene-modified” to “transgenic”
Line 37 Correct to “Cry protoxins (~130 or ~75 kDa) are produced…”
Line 39 Correct to (~60 kDa)
Lines 79, 98 and elsewhere: the authors refer to “the Heckel group”. This is not appropriate, as the work cited was performed by multiple laboratories and researchers that were not directed by Heckel (he was just one of the contributors and Linda Gahan, Fred Gould and others directed their own groups to do the work). Please substitute for “Gahan et al”.
Line 138: Authors state that receptors are needed to form pores efficiently. Another possibility is that pores in the presence of receptors are “different” (have distinct properties) than the ones formed in lipid bilayers.
Line 139: This statement of gut fluid being three dimensional and gut cell membrane two dimensional is incorrect. The cell membrane is a three-dimensional structure too! Please clarify or correct accordingly.
Line 145: “receptors must bind to the toxin” is incorrect, as the receptor is localized to the gut epithelium and the toxin is in the mobile phase (gut fluids) and able to seek and bind receptors. Should be “toxin must bind to receptors”.
Lines 196 and 200 contain the same statement (close to the domain I/II boundary), so it reads repetitive.
Fig. 1- The concept of Kdmut/Kdwt may be difficult to understand for non-specialists, and in the current version is only defined in the figure legend. I suggest explaining in the text what a higher or lower value means regarding the importance of that amino acid for binding.
Fig. 2-The authors state a correlation between susceptibility and Kd of receptors. However, all the ABCC2 examples in the graph show the same Kd but different toxicity depending on the Cry protein. This suggests that affinity is not the only contributor to pore formation efficacy, and that the toxin is also important. For example, it may be that distinct toxins bind to different regions and this may affect the pore formation activity. Authors should comment on this.
Line 242: Define BmCad-TBR
Fig. 3C: Were there no mutations affecting only PEBmCad?
Fig. 4 and section 2.2.4: Authors try to explain effects of ABC and cadherin receptor knockouts, but they did not discuss the possibility of mutations in ABCC3. It would also be nice to have specific examples underneath each of the cases/panels. For instance, Cry1F resistance in fall armyworm as example of B.
Line 288: Lower affinity compared to what?
Line 304: The observation that Cry1Ac toxicity in transgenic Drosophila expressing APN is not does-dependent is very interesting. This would suggest that high levels of activity are only observed when enough toxin is present for APN to facilitate pore formation. Or APN may accumulate enough toxin at that point on the membrane so that any internal cadherin or ABC protein can facilitate pore formation. I think it is OK for authors to speculate on these possibilities.
Line 329: References are needed here.
Fig. 5 and section 2.3.2: The authors discuss contradictory results reported from genetic knockout strains of Plutella, but it is not clear they provide a clear explanation for these contradictory results. How does their model explain low resistance in Liu et al or Zhao et al but high resistance in Guo et al in ABCC2 knockouts?
Line 359: Authors explain binding to receptors is needed for oligomerization. However, Masson et al (Biochim Biophys Acta 2002 Vol. 1594 Issue 2 Pages 266-75) Observed the formation of multimeric Cry structures without interaction with receptors. Is it possible these oligomers differ from the ones observed in the presence of receptors? This receptor-independent multimerization may affect some of the stated assumptions and authors should discuss this.
Fig. 7: From a visual standpoint, it is unclear why the monomer toxin after the conformational change upon binding to cadherin and ABCC2 differs in probability for membrane insertion. In both cases domain I is close to the membrane and the rest of the toxin remains bound to the receptor. In the text, authors argue it may have to do with domain I distance from the membrane, but this is not visible in the figure. I suggest scaling down the toxin molecule to a closer relative size in respect to the receptor in displaying relative distance to the membrane in each case (I assume drawings are not to scale).
Line 513: The authors claim “Given that…” but I think this is an assumption (unless they provide supportive citations).
“Synergism” is discussed throughout the paper but not clearly defined. Some of the effects (Ex. Fig. 10A) seem to be additive rather than synergistic. Is “synergism” used instead of “enhancement” or as by definition (effect is much higher than expected from additive interactions). It would help to define early in the manuscript, for the sake of clarity.
Fig. 11 is vital to the formula provided predicting toxicity (correct “actual” in the legend to “observed”). The figure presents a correlation between the observed and predicted toxicity using the formula proposed by the authors. However, the data in Fig. 10 and elsewhere in the paper support that ABCC2 is clearly a better receptor for pore formation than ABCC3. Consequently, one would expect that cells expressing ABCC3/cad would be less sensitive to any Cry protein compared to cells expressing ABCC2/cad (as shown in the figure). Therefore, this correlation could have been guessed without the use of any formulas. More predictive would be to find a correlation between toxicity in cells expressing ABCC2/cad when different Cry proteins are used. This is not the case, as toxicity in C2/Cad-1Ab and C2/Cad-1Ac seems much higher than what is predicted by the formula (two orders of magnitude). Consequently, the relevance of the proposed formula is unclear.
Line 576-577: Considering the issues stated above (additional receptors, formula not being accurate predictor), this statement is incorrect and should be deleted.
Line 584: Cry-resistant strains do not express functional receptors. How would then the proposed model based on two receptors commonly mutated in resistant insects help design Cry proteins overcoming resistance? Cry1Mod toxins overcame resistance, but the mechanism involved is unclear and may not be receptor-mediated.
Author Response
This review summarizes knowledge on the binding and toxicity (as a proxy to pore formation) of Cry1A proteins in Lepidoptera. The authors also provide a hypothetical model combining published data supporting that binding kinetics directly predict pore formation and receptor activity, and that the kinetics of distinct receptors explain cases of synergism. While the idea of binding kinetics being directly related to pore formation is well established in the field, the hypothesis of synergism being dependent on binding kinetics of receptors involved seems novel and supported by the data presented.
>Thank you very much for your time to review our manuscript and provide many important suggestions. As you mentioned, it has been widely accepted in the field that binding affinity directly affects pore formation, which is supported by evidence that Cry binding affinity to BBMV largely correlates its toxicity (e.g. Masson et al., 1995 JBC). Meanwhile, binding affinity to a certain Cry-binding protein (cadherin, APN…) could not explain differences in toxicity. Here we report a strong correlation between Cry binding affinity to ABC transporters and ABC-mediated toxicity. Furthermore, the synergism is also dependent on pore formation efficiency of ABC transporters. We consider this is one of the highlights of this article.
Since the review and proposed model is built on data with Cry1 proteins and Bombyx mori, I think the title should be revised to be more specific. As written, the current tile is too general and is misleading as the review does not go into detail into other Cry proteins. Suggested title: “Molecular and kinetic model for Cry1 protein pore formation in silkworm (Bombyx mori)”. This reviewer has other relevant points and minor corrections for the authors to consider.
>We do not feel that the current title is too general. The models we propose here and principles we are finding are not specific to the combination of Cry1 and Bombyx but could generally apply to any combinations of a Cry toxin and a receptor (at least ABC transporters and cadherin). For example, Cry2 and Cry3 binding affinity to ABCA2 and ABCB1 should directly correlate with ABCA2- and ABCB1-mediated toxicity. Even if novel receptors and novel synergisms will be discovered, we believe that both of our molecular and kinetic models have potential to explain the mechanisms with minor modification.
One of the main points is that alkaline phosphatases are recognized as receptors in the literature akin to APNs, but are not considered or mentioned in the review. I recognize there is not much data available, but I consider it deserves to be mentioned as published reports support it is a relevant binding site for toxicity of Cry proteins.
>According to the suggestion, we discuss ALP in the section 2.3. as well as APN.
I also have concerns with the model proposed to explain cadherin/ABCC2 synergism and its biological relevance, especially since authors boldly propose that the formulas in 573-577 can be used to predict insect susceptibility. The biggest concern is that the proposed model is entirely built on data from in vitro experiments with cultured cells expressing B. mori receptors. This is an artificial system that may be used to explore some questions, but is probbaly not amenable to explain toxicity in vivo.
>While we understand what the reviewer concerns, we believe that data from in vitro experiments theoretically explain in vivo susceptibility for following two reasons. First, the killing mechanism of Cry protein is the same both in in vitro and in vivo across insect species. Second, the killing mechanism of Cry1A in vivo generally depends on ABCC2 and cadherin because the lack of them causes high resistance.
There are a number of other proteins in the gut known to interact with Cry proteins that would affect the outcomes of the interactions modeled here, and these proteins are not considered in the model (ALP, APN, lipophorin…). For instance, the Cry oligomer bound to cadherin displays high affinity binding to APN and ALP. This would limit the transfer of oligomer from cadherin to ABCC2 proposed in Fig. 9.
>As you pointed out, much more interactions must exist in vivo than in vitro. However, do such interactions (e.g. oligomer binding to ALP and APN) occur in vivo and affect Cry toxicity? At least, oligomer binding to ALP and APN was suggested only by in vitro experiments using artificially prepared Cry oligomers. Furthermore, high resistance by ABCC2 indicates that, even if such interactions would exist, the interaction with ABCC2 governs Cry1A susceptibility in vivo. So, when discussing the primary susceptibility determinant or the dominant killing mechanism, probably we do not need to pay much attention for the interactions you mentioned. Even when the expression level of APN is 100-fold higher than that of ABCC2, the Cry binding amount is almost the same considering their binding affinities and this causes just two-fold differences in Cry binding to ABCC2.
Overlapping Cry binding sites in cadherin and ABCC2 also present an issue for the model. If binding to ABCC2 is more conducive to pore formation (toxicity), then less productive Cry binding to cadherin would reduce the number of pores formed compared to when ABCC2 is the only receptor (again, this is a limitation of a model based on invitro tests with one or two receptors present). Synergism in this system of two receptors would only be observed when binding sites are saturated or when the affinity of the less favorable receptor (cadherin) is lower (higher Kd) than the more effective receptor (ABCC2). This is in contrast to what authors claim in line 530.
>In vitro experiments showing the ABCC2-cadherin synergism suggested that the overlapping binding sites do not limit the toxicity but rather, in such a condition, cadherin-binding Cry protein can participate in pore formation more efficiently than when expressing cadherin alone. What you concern may happen only when cadherin cannot mediate oligomerization and pore formation (cannot act as a receptor) but binds to Cry with high affinity. To our knowledge, such cadherin has never reported so far. Even if such cadherin would exist, it does not dramatically affect toxicity unless the expression level of cadherin is extremely high than that of ABCC2.
In any case, for this system to work the Cry protein must be released from ABCC2 so the receptor is recycled. More importantly, other gut proteins surely would affect this process and are not considered in the model. Authors should consider these limitations and include them in the text.
>It is unclear whether release of Cry protein and recycle of receptors are needed. In that regard, we referred to the possibility of dissociation of Cry after membrane insertion (lines 380-383). There is no evidence supporting that other gut proteins affect the release of Cry protein from ABCC2 and cadherin and recycle of the two receptors.
While the paper is well organized, it is difficult to follow at times due to the style of writing. There are also some typos that need to be corrected for clarity. An important correction is that species names should be always italicized (none of the names are italicized in the text of the version I reviewed).
>We feel sorry for the inconvenience. We have corrected typos and species names as far as possible.
The authors use “toxin” sometimes referring to Cry1A toxins or Cry toxins, so unless they refer to a specific protein, they should use the plural (Ex: line 36 “Cry toxin is classified…” should be “Cry toxins are classified…”. Also, I would suggest using “insecticidal protein” or “pesticidal protein” instead of “toxin”.
>Thank you for the suggestion. We revised to use the plural and changed “Cry toxin” to “Cry protein” along the text.
Line 32 “moths” are not larvae (ok to state beetle and mosquito larvae). Change to “caterpillars”.
>We changed “moths” to “caterpillars”.
Line 33 “Cry genes” should be “Cry proteins”.
>We refer Cry genes as a gene source for insect-resistant transgenic crops, so “Cry genes” are correct.
Line 34 Correct “gene-modified” to “transgenic”
>We changed “gene-modified” to “transgenic”.
Line 37 Correct to “Cry protoxins (~130 or ~75 kDa) are produced…”
>kDa is often used to indicate molecular weight and we used it before but we have learned that it is incorrect scientifically because molecular weight is a dimensionless quantity. Here we use kDa as a unit for mass (m/kDa means mass/kDa).
Line 39 Correct to (~60 kDa)
>See the response to the previous comment.
Lines 79, 98 and elsewhere: the authors refer to “the Heckel group”. This is not appropriate, as the work cited was performed by multiple laboratories and researchers that were not directed by Heckel (he was just one of the contributors and Linda Gahan, Fred Gould and others directed their own groups to do the work). Please substitute for “Gahan et al”.
>According to your suggestion, we substitute “the Heckel group” for “Gahan et al.” (lines 81 and 99). Also, we changed “the Bravo and Soberon group” to “Bravo et al.” (line 90).
Line 138: Authors state that receptors are needed to form pores efficiently. Another possibility is that pores in the presence of receptors are “different” (have distinct properties) than the ones formed in lipid bilayers.
>According to Schwartz et al. (J. Membrane Biol. 132, 53-62, 1993), permeation and kinetic properties of Cry1C pores in receptor-free artificial lipid bilayer are similar to those of pores in Sf9 cells endogenously expressing unknown Cry1C receptor. On the other hand, when partially-purified receptor complex from M. sexta BBMV was reconstituted in planar lipid bilayers, Cry1Ac formed pores at ~250-fold lower concentration (0.33 nM) (Schwartz et al., FEBS Letters 412, 270-276, 1997). These suggest that receptors contribute to efficiency of pore formation but not to properties of pores.
Line 139: This statement of gut fluid being three dimensional and gut cell membrane two dimensional is incorrect. The cell membrane is a three-dimensional structure too! Please clarify or correct accordingly.
>Yes, exactly. We changed it to “two-dimensional (the surface of target cell membrane)” (line 145) to clarify the meaning.
Line 145: “receptors must bind to the toxin” is incorrect, as the receptor is localized to the gut epithelium and the toxin is in the mobile phase (gut fluids) and able to seek and bind receptors. Should be “toxin must bind to receptors”.
>According to the suggestion, we revised the sentence to “For efficient pore formation, domain I of Cry protein must be partially unfolded by binding to a receptor near the cell membrane” (line 150-152).
Lines 196 and 200 contain the same statement (close to the domain I/II boundary), so it reads repetitive.
>Thanks for pointing it out. Here we would like to mention that the ABCC2-binding site is close to the two salt bridges while cadherin-binding site is close to just one. We changed the sentence as follows: “Thus, the ABCC2-binding site on Cry1A is close to the two domain I–II salt bridges.”(lines 205 and 206)
Fig. 1- The concept of Kdmut/Kdwt may be difficult to understand for non-specialists, and in the current version is only defined in the figure legend. I suggest explaining in the text what a higher or lower value means regarding the importance of that amino acid for binding.
>Thanks to your comment, we found Kdmut/Kdwt was a mistake and Kdwt/Kdmut is correct and revised it. We added explanation “High and low values indicate high and low binding affinity to BmABCC2 and BmCad” in the legend because in the text readers do not need to interpret the data on their own.
Fig. 2-The authors state a correlation between susceptibility and Kd of receptors. However, all the ABCC2 examples in the graph show the same Kd but different toxicity depending on the Cry protein. This suggests that affinity is not the only contributor to pore formation efficacy, and that the toxin is also important. For example, it may be that distinct toxins bind to different regions and this may affect the pore formation activity. Authors should comment on this.
>Yes, we agree with the comment. We guess that the reviewer might just overlook our explanation in lines 219-223 which we had mentioned your point. And in that context, we defined and introduced “pore formation efficiency” in the formula as another factor affecting ABC-mediated toxicity.
Line 242: Define BmCad-TBR
>We added explanation “BmCad-TBR (TBR, toxin-binding region)”.
Fig. 3C: Were there no mutations affecting only PEBmCad?
>Yes, as far as tested mutants shown in Fig. 3C, no mutants specifically affected PEBmCad.
Fig. 4 and section 2.2.4: Authors try to explain effects of ABC and cadherin receptor knockouts, but they did not discuss the possibility of mutations in ABCC3. It would also be nice to have specific examples underneath each of the cases/panels. For instance, Cry1F resistance in fall armyworm as example of B.
>In Fig. 4, we attempt to explain typical mechanisms of resistance. It is difficult to refer to specific examples because in many cases no parallel data are available (susceptibility of receptor-KO larvae and in vitro evaluation of contributions of pore formation by each receptor candidates). We discuss an example of ABCC3 mutation (almost equal contributions of ABCC2 and ABCC3) in Fig. 5 (left panel).
Line 288: Lower affinity compared to what?
> We added the explanation “in comparison to ABCC2 and cadherin (KD = ~0.1-1 nM)” (lines 295 and 296).
Line 304: The observation that Cry1Ac toxicity in transgenic Drosophila expressing APN is not does-dependent is very interesting. This would suggest that high levels of activity are only observed when enough toxin is present for APN to facilitate pore formation. Or APN may accumulate enough toxin at that point on the membrane so that any internal cadherin or ABC protein can facilitate pore formation. I think it is OK for authors to speculate on these possibilities.
>We do not agree with that idea. It is reasonable to think that the dose-independent observation may be because the experiment did not work properly due to unknown reason(s).
Line 329: References are needed here.
>Thank you for pointing it out. We added references in line 338.
Fig. 5 and section 2.3.2: The authors discuss contradictory results reported from genetic knockout strains of Plutella, but it is not clear they provide a clear explanation for these contradictory results. How does their model explain low resistance in Liu et al or Zhao et al but high resistance in Guo et al in ABCC2 knockouts?
>If both results are true, we cannot explain what is happening. Our interpretation is that results from Liu et al. and Zhao et al. can be explained easily based on functional redundancy of ABCC2 and ABCC3, but those from Guo et al. require multiple unknown mechanisms to reasonably interpret. To clarify that, we added following explanation in lines 336-342: “Functional redundancy of the two ABC transporters in Cry1Ac and Cry1F susceptibility were reported in other lepidopteran species [38,41]. Thus, the results from Liu et al. and Zhao et al. can be reasonably interpreted based on known mechanisms. In contrast, the results from Guo et al. suggest that different mechanisms based on putative unknown “synergisms” underlie the primary Cry1Ac susceptibility determinant in the same species. “
Line 359: Authors explain binding to receptors is needed for oligomerization. However, Masson et al (Biochim Biophys Acta 2002 Vol. 1594 Issue 2 Pages 266-75) Observed the formation of multimeric Cry structures without interaction with receptors. Is it possible these oligomers differ from the ones observed in the presence of receptors? This receptor-independent multimerization may affect some of the stated assumptions and authors should discuss this.
> Masson et al. observed spontaneus oligomerization in low salt conditions (0 and 50 mM) while the K+ concentration in lepidopteran midgut digestive fluids is high (~300 mM). So, probably we do not need to pay much attention to the receptor-independent oligomerization. Regarding the structural differences of oligomers with and without receptors, no precise information is available so far. However, as mentioned above, permeation and kinetic properties of Cry1C pores were similar between receptor-free and receptor-expressing membrane (Schwartz et al., 1993). Their structures may be identical. Because it is obvious that Cry monomer is dominant when binding to a receptor, we think that we do not need to mention it in the text.
Fig. 7: From a visual standpoint, it is unclear why the monomer toxin after the conformational change upon binding to cadherin and ABCC2 differs in probability for membrane insertion. In both cases domain I is close to the membrane and the rest of the toxin remains bound to the receptor. In the text, authors argue it may have to do with domain I distance from the membrane, but this is not visible in the figure. I suggest scaling down the toxin molecule to a closer relative size in respect to the receptor in displaying relative distance to the membrane in each case (I assume drawings are not to scale).
>We understand your point. Because cadherin mediates Cry1A pore formation by itself with low efficiency and via the synergism with ABCC2 with high efficiency, helices of cadherin-binding Cry1A should manage to insert the cell membrane. But on the other hand, the efficiency is much lower than that of ABCC2. So, it is difficult to express the nuance in the drawings. The main point in Fig. 7 is that most Cry-ABC complexes form pores while most Cry-cadherin complexes form prepores. By the way, the drawings in Fig. 7 were roughly scaled. Regarding the distance from the membrane, we think that Fig. 8 is helpful to see it.
Line 513: The authors claim “Given that…” but I think this is an assumption (unless they provide supportive citations).
>As the reviewer pointed out, this is an assumption. We changed the phrase to “on the assumption that…” (line 534).
“Synergism” is discussed throughout the paper but not clearly defined. Some of the effects (Ex. Fig. 10A) seem to be additive rather than synergistic. Is “synergism” used instead of “enhancement” or as by definition (effect is much higher than expected from additive interactions). It would help to define early in the manuscript, for the sake of clarity.
>We use the term “synergism” as literal meaning though the text. In Fig. 10A, the amount of binding toxin is “additive” but the toxicity is “synergistic”. Cadherin can mediate 10% cell swelling at high Cry concentration (~300-1000 nM) but Cry binds to cadherin in much lower concentration (0.1~1nM) because of its high binding affinity. In this paper we propose that the insertion of cadherin-binding Cry by ABC-binding Cry enables highly efficient pore formation in lower Cry concentration. Fig. 10C explains an example of no synergism (additive) because of low Cry1F binding affinity of cadherin. We added the explanation of the synergism in lines 114-119.
Fig. 11 is vital to the formula provided predicting toxicity (correct “actual” in the legend to “observed”). The figure presents a correlation between the observed and predicted toxicity using the formula proposed by the authors. However, the data in Fig. 10 and elsewhere in the paper support that ABCC2 is clearly a better receptor for pore formation than ABCC3. Consequently, one would expect that cells expressing ABCC3/cad would be less sensitive to any Cry protein compared to cells expressing ABCC2/cad (as shown in the figure). Therefore, this correlation could have been guessed without the use of any formulas. More predictive would be to find a correlation between toxicity in cells expressing ABCC2/cad when different Cry proteins are used. This is not the case, as toxicity in C2/Cad-1Ab and C2/Cad-1Ac seems much higher than what is predicted by the formula (two orders of magnitude). Consequently, the relevance of the proposed formula is unclear.
>We feel sorry that lack of some explanations might bother the reviewer. The magnitude of the synergistic effect varies across ABC-cad combinations and across Cry proteins. For instance, The BmABCC2-BmCad synergism causes 10-, 1000-, and 100-fold higher toxicity of Cry1Aa, Cry1Ab, and Cry1Ac when compared to BmABCC2 alone and no synergism was observed in Cry1Fa. In Fig. 11, we tested whether our formula could explain the difference in the magnitude and predict the synergism-mediated toxicity. The strong correlation supported the formula, although we acknowledge the prediction for C2/Cad-1Ab is somehow wrong. We added the explanation in the text (lines 523-526).
Line 576-577: Considering the issues stated above (additional receptors, formula not being accurate predictor), this statement is incorrect and should be deleted.
>As we explained in the response to the previous comment, we believe that the formula is useful as a prototype to predict Cry toxicity. For accurate explanation, we revised the sentence as follows: “In vivo toxicity toward pest individuals will be predicted if a more accurate mathematical model incorporating the expression levels of receptors and other factors is built using our formulas as prototypes” (lines 601-603).
Line 584: Cry-resistant strains do not express functional receptors. How would then the proposed model based on two receptors commonly mutated in resistant insects help design Cry proteins overcoming resistance? Cry1Mod toxins overcame resistance, but the mechanism involved is unclear and may not be receptor-mediated.
>For example, as we explained in lines 627-629, protein engineering will enable to confer high binding affinity to other ABC transporters expressed in midgut and some obtained mutants could have potential to be active to the resistant strains. Furthermore, it is better to understand the putative receptor-independent mechanism of Cry1Mod. In this section, we discuss the general advantage of a complete understanding of pore formation mechanism in overcoming resistance, including beyond our model. Our models provide some strategies but other strategies will be provided by receptor-independent mechanism and other novel mechanisms.
Reviewer 3 Report
This is a well written manuscript, summarising present knowledge on pore formation mechanism on epithelial cells targeted by Cry toxins. The authors present also their model, critically addressing what has been known so far and what still remains to be understood.
Author Response
Thank you for your kind reviewing our manuscript.
Round 2
Reviewer 2 Report
I appreciate the authors' efforts to provide clear explanations that helped resolved most of the concerns raised to the previous version, but some relevant issues remain. I also appreciate the author's efforts in contributing potential models explaining synergism between receptors. However, this reviewer still considers (as explained below) that the model is developed using a very limited dataset of receptors from one insect and Cry1A-1F toxins, does not consider other receptors present in the insect that would affect toxicity, and even some of the predictions (Cry1Ab and Cry1Ac with C2/cad) are very inaccurate. It is valuable to do the exercise of developing a model, but authors should keep in mind and acknowledge the above limitations, for example by seriously toning down their claims about the model being all-encompassing and being useful for predictions in vivo. Unless more data are presented, the model as proposed is probably limited to the system used to develop it and would be risky to assume it applies to other toxin/receptors or in vivo.
Specific responses to comments:
"We do not feel that the current title is too general. The models we propose here and principles we are finding are not specific to the combination of Cry1 and Bombyx but could generally apply to any combinations of a Cry toxin and a receptor (at least ABC transporters and cadherin). For example, Cry2 and Cry3 binding affinity to ABCA2 and ABCB1 should directly correlate with ABCA2- and ABCB1-mediated toxicity. Even if novel receptors and novel synergisms will be discovered, we believe that both of our molecular and kinetic models have potential to explain the mechanisms with minor modification."
As stated in my review, association between binding affinity to receptors (especially ABCC2) and toxicity is not new. The novelty of the model is trying to explain synergisms. It is plausible that the model may help explain toxicity synergism of between ABC and cadherins in Cry2 and Cry3 (or other Cry1 proteins), but in the absence of these data I think it is misleading to use a title that suggests a model that explains synergism for all Cry proteins.
"While we understand what the reviewer concerns, we believe that data from in vitro experiments theoretically explain in vivo susceptibility for following two reasons. First, the killing mechanism of Cry protein is the same both in in vitro and in vivo across insect species. Second, the killing mechanism of Cry1A in vivo generally depends on ABCC2 and cadherin because the lack of them causes high resistance."
I disagree with the authors. While it is assumed, there is no clear data supporting that the killing mechanism in vitro and in vivo is the same, especially considering that in vitro cell cultures differ from mature midgut cells targeted in vivo. Thus, the model develop has to be taken with caution and the bold claim by the authors that can be used to explain and predict in vivo toxicity is not supported (and should be deleted) unless actual data to this point are presented. In addition, on the second point, there is at least one clear example in which high levels of resistance are only observed when a mutation in ABCC2 coexist with reduced APN1 levels (Ex: Ma et al, Insect Biochem Mol Biol 2022). This is clear evidence that the role of APN and ALP should be considered in any attempt at a complete model. I agree ABCC2 and cadherin seem critical in most cases, but other proteins seem to also have a relevant effect on susceptibility and are not considered in the model. At the very least, this limitation should be recognized in the text.
"On the other hand, when partially-purified receptor complex from M. sexta BBMV was reconstituted in planar lipid bilayers, Cry1Ac formed pores at ~250-fold lower concentration (0.33 nM) (Schwartz et al., FEBS Letters 412, 270-276, 1997). These suggest that receptors contribute to efficiency of pore formation but not to properties of pores."
In the study of Schwartz et al (1997) brought up by the authors, they used GPI-enriched BBMV proteins, which would contain high levels of APN and ALP but no cadherin or ABCC2. The pore formation observed in that case also supports a role for APN and ALP that is not considered in the model proposed by the authors. Moreover, in the same paper by Schwrartz et al it is discussed that properties of the pores differ when the BBMV proteins are present, in line with the argument I brought up stating that properties of the pores may depend on variables of the system (as lipid composition or cell type). Therefore, pore properties in cultured cells may differ from midgut cells, which makes it difficult to expand the proposed model to in vivo systems. The authors should recognize this limitations clearly in the paper. As stated before, unless additional data with other toxins/receptors are produced, the model is right now limited to silkworm ABCC2 and cadherin (thus my request for a more specific title and toning down claims on extending the model to in vivo conditions).
"Yes, exactly. We changed it to “two-dimensional (the surface of target cell membrane)” (line 145) to clarify the meaning."
I am still missing the point here. The surface of the target cell is a 3-dimensional surface, not 2D. I strongly suggest rewording for clarity.
"In Fig. 4, we attempt to explain typical mechanisms of resistance. It is difficult to refer to specific examples because in many cases no parallel data are available (susceptibility of receptor-KO larvae and in vitro evaluation of contributions of pore formation by each receptor candidates). We discuss an example of ABCC3 mutation (almost equal contributions of ABCC2 and ABCC3) in Fig. 5 (left panel)."
I did not make my point clear, apologies. My suggestion was referring to cases of resistance when the mutation involved is known (for instance ABCC2 mutations in fall armyworm, cadherins in pink bollworm...).
"We do not agree with that idea. It is reasonable to think that the dose-independent observation may be because the experiment did not work properly due to unknown reason(s)"
I respect if the authors would rather not comment on the Drosophila-APN observation, but I strongly disagree with their rationale. Results from experiments tell us something, so the experiments always work. If unexpected results are observed we should come up with hypothesis to explain them, rather than discard them as not working properly.
"However, as mentioned above, permeation and kinetic properties of Cry1C pores were similar between receptor-free and receptor-expressing membrane (Schwartz et al., 1993). Their structures may be identical. Because it is obvious that Cry monomer is dominant when binding to a receptor, we think that we do not need to mention it in the text."
As stated above, in the later publication (Schwartz et al 1997), it is described that pore properties could differ in the absence/presence of receptors (in that case APN/ALP). This suggests pores may differ. Also, Cry monomer being dominant when binding to receptor is not completely accurate. Cry oligomers display much higher affinity for APN/ALP, so that would be the dominant form binding to these proteins. I strongly suggest authors include these observations in the text.
"The BmABCC2-BmCad synergism causes 10-, 1000-, and 100-fold higher toxicity of Cry1Aa, Cry1Ab, and Cry1Ac when compared to BmABCC2 alone and no synergism was observed in Cry1Fa. In Fig. 11, we tested whether our formula could explain the difference in the magnitude and predict the synergism-mediated toxicity. The strong correlation supported the formula, although we acknowledge the prediction for C2/Cad-1Ab is somehow wrong. We added the explanation in the text (lines 523-526)."
There is no mention or explanation of Cry1Ab not following the linear relationship in the text. Also, one would argue that the model also does not work well for Cry1Ac, as the observed toxicity is an order of magnitude lower than the predicted. These observations suggest the model does not accurately predict toxicity for 2 out of 3 C2/Cad-Cry1A toxins tested. Differences of an order of magnitude in an in vitro system may translate to important inconsistencies between predicted and observed toxicity in vivo, throwing more doubt on the validity of the model. Strongly suggest to changing "actual" to "observed" toxicity.
"As we explained in the response to the previous comment, we believe that the formula is useful as a prototype to predict Cry toxicity. For accurate explanation, we revised the sentence as follows: “In vivo toxicity toward pest individuals will be predicted if a more accurate mathematical model incorporating the expression levels of receptors and other factors is built using our formulas as prototypes” (lines 601-603)."
This comment goes back to the observation that the model is developed using in vitro data with a limited set of Cry receptors (Bmcadherin and BmABCC2) and toxins (Cry1A). Additional data with other toxins or insect receptors is needed to support the idea that the model is useful for in vivo predictions. Optimally, one would estimate synergisms with the formula first and then test in vitro to confirm the predictions. Without these data, I do not think the current paper provides enough information to support how applicable the model is to other toxin/insect systems, especially in vivo.
Author Response
I appreciate the authors' efforts to provide clear explanations that helped resolved most of the concerns raised to the previous version, but some relevant issues remain. I also appreciate the author's efforts in contributing potential models explaining synergism between receptors. However, this reviewer still considers (as explained below) that the model is developed using a very limited dataset of receptors from one insect and Cry1A-1F toxins, does not consider other receptors present in the insect that would affect toxicity, and even some of the predictions (Cry1Ab and Cry1Ac with C2/cad) are very inaccurate. It is valuable to do the exercise of developing a model, but authors should keep in mind and acknowledge the above limitations, for example by seriously toning down their claims about the model being all-encompassing and being useful for predictions in vivo. Unless more data are presented, the model as proposed is probably limited to the system used to develop it and would be risky to assume it applies to other toxin/receptors or in vivo.
>Thank you very much for your kind reviewing our manuscript again. We feel sorry if there was our lack of clarity. We would like to share afresh our basic view for the kinetic model with the reviewer. First, as a matter of course, we do not consider that the model predicts everything of in vivo toxicity and did not argue it in the text. As binding affinity and pore formation efficiency seem to largely explain in vitro toxicity mediated by ABCC2 and cadherin, we believe it is useful as a “prototype” for building a much more accurate model. We just state the potential of our model. Second, the reviewer concerns about limited dataset to develop the model. We used seven combinations two ABC transporters and four Cry proteins, which mediate high and low toxicity, to seek the universal kinetic principle underlying Cry toxicity. Although additional dataset is welcome, we think that data used is not a very limited dataset but enough to provide plausible idea. Third, the reviewer consider that more data is needed. We totally agree that experimental verifications are essential to test the model using various combinations of Cry proteins and their receptors from many insect species. However, what we discuss here is a posteriori reasoning and we discuss the potential of the model to explain the basic principle underlying Cry toxicity. Please note that we discuss the potential of our model to explain in vivo toxicity in just one sentence (lines 600-603).
Specific responses to comments:
As stated in my review, association between binding affinity to receptors (especially ABCC2) and toxicity is not new. The novelty of the model is trying to explain synergisms.
>Although we may need to learn from your review, we feel that what you refer here is somehow different from what we consider new. Anyway, we are happy if there is a consensus in the field about the correlation between Cry binding affinity to ABC and ABC-mediated toxicity.
It is plausible that the model may help explain toxicity synergism of between ABC and cadherins in Cry2 and Cry3 (or other Cry1 proteins), but in the absence of these data I think it is misleading to use a title that suggests a model that explains synergism for all Cry proteins.
>Here we propose “models” of Cry pore formation using current insights on Cry1 proteins and ABCC2/Cadherin. We guess the reviewer agree with that Cry proteins have the conserved mode of action. Actually, ABCA2 and ABCB1 are receptors for Cry2 and Cry3. It is natural to think what we have learned from Cry1A-ABCC2 could apply to their mode of action. We do not refer to synergisms in Cry2 and Cry3 mode of action because cadherin is not likely their receptors. Furthermore, the synergism is one of the topics in this paper.
I disagree with the authors. While it is assumed, there is no clear data supporting that the killing mechanism in vitro and in vivo is the same, especially considering that in vitro cell cultures differ from mature midgut cells targeted in vivo. Thus, the model develop has to be taken with caution and the bold claim by the authors that can be used to explain and predict in vivo toxicity is not supported (and should be deleted) unless actual data to this point are presented.
>Again, we do not state that the formula is directly useful to predict in vivo toxicity. As the reviewer pointed out, more research is needed to say that the killing mechanism is “completely” the same. However, it is plausible that the primary toxicity is mediated by pore formation and colloid-osmotic lysis, which is mediated by ABC transporters and cadherin (plus unknown receptors and minor effects by other proteins such as APN and ALP). To clarify the meaning, we revised the sentence in lines 601-604 as follows: “In the future, in vivo toxicity toward pest individuals may be predicted if a much more accurate mathematical model is built using our formulas as prototypes by incorporating contributions to other receptors (if exist), the expression levels of receptors, and other factors filling gaps between in vivo and in vitro”.
In addition, on the second point, there is at least one clear example in which high levels of resistance are only observed when a mutation in ABCC2 coexist with reduced APN1 levels (Ex: Ma et al, Insect Biochem Mol Biol 2022). This is clear evidence that the role of APN and ALP should be considered in any attempt at a complete model. I agree ABCC2 and cadherin seem critical in most cases, but other proteins seem to also have a relevant effect on susceptibility and are not considered in the model. At the very least, this limitation should be recognized in the text.
>Ma et al. wrote “Whether and how the APN1 or APN6 directly involves in the toxicity of, or resistance to, Cry1Ac in T. ni requires to be understood” in the last sentence of Discussion. So, the interpretation of this paper by the reviewer seems not accurate. So far, down- and up-regulation of APN1 and APN6 are just correlations observed in the resistant strain and causal effects have not been demonstrated. From available data at this stage, we think that APN and ALP do not make an important contribution to pore formation. At the same time, however, we do not rule out any other mechanisms including by other proteins.
In the study of Schwartz et al (1997) brought up by the authors, they used GPI-enriched BBMV proteins, which would contain high levels of APN and ALP but no cadherin or ABCC2. The pore formation observed in that case also supports a role for APN and ALP that is not considered in the model proposed by the authors.
> Note that BBMV was partially purified by gel filtration in the paper by Schwartz et al., but it contained many other proteins in addition to APN and ALP. So, results from the paper cannot support a role of a certain protein. Even if APN and ALP contribute to pore formation observed in the study, as the patch-clamp recording is very highly sensitive method to detect channel activity, it is uncertain whether the pore formation by APN and ALP could contribute to cell death.
Moreover, in the same paper by Schwrartz et al it is discussed that properties of the pores differ when the BBMV proteins are present, in line with the argument I brought up stating that properties of the pores may depend on variables of the system (as lipid composition or cell type). Therefore, pore properties in cultured cells may differ from midgut cells, which makes it difficult to expand the proposed model to in vivo systems. The authors should recognize this limitations clearly in the paper. As stated before, unless additional data with other toxins/receptors are produced, the model is right now limited to silkworm ABCC2 and cadherin (thus my request for a more specific title and toning down claims on extending the model to in vivo conditions).
>As the reviewer stated, Schwartz et al. discussed that the difference may be related to lipid composition. But this is a difference in the artificial lipid bilayer. Unless the midgut columnar cells have extraordinary lipid composition, we can reasonably assume that pores mediated by the same receptor in insect cell lines and midgut cells have the same property.
I am still missing the point here. The surface of the target cell is a 3-dimensional surface, not 2D. I strongly suggest rewording for clarity.
>We deleted “two- and three-dimensional” and changed to “The action of binding to Cry proteins and gathering them from gut fluid to the surface of target cell membrane is required for efficient interactions to form pores” (lines 144-145).
I did not make my point clear, apologies. My suggestion was referring to cases of resistance when the mutation involved is known (for instance ABCC2 mutations in fall armyworm, cadherins in pink bollworm…).
>We agree that it is ideal, but it would be misleading because the contributions of ABCC2, ABCC3, cadherin and other proteins are different among species. For example, resistance by cadherin mutation does not always suggest a synergism with ABCC2 but also ABCC3. Because we do not guarantee the accuracy and Fig. 4 is just a schematic example, we think it is better not to refer specific examples in this paper (your review could be useful for considering that).
I respect if the authors would rather not comment on the Drosophila-APN observation, but I strongly disagree with their rationale. Results from experiments tell us something, so the experiments always work. If unexpected results are observed we should come up with hypothesis to explain them, rather than discard them as not working properly.
>We understand what the reviewer says but it is also true that things suggested by one paper do not directly mean that is truth. Actually, dose-independent mode of action mediated by APN is not reported elsewhere. What we want to say here is that this is not strong evidence. We do not just discard the result but suggest additional verifications using other methods to clarify the role (lines 318-320).
As stated above, in the later publication (Schwartz et al 1997), it is described that pore properties could differ in the absence/presence of receptors (in that case APN/ALP). This suggests pores may differ. Also, Cry monomer being dominant when binding to receptor is not completely accurate. Cry oligomers display much higher affinity for APN/ALP, so that would be the dominant form binding to these proteins. I strongly suggest authors include these observations in the text.
>As state above, the difference in properties was observed in the artificial lipid bilayer. We do not rule out the possibility in the text. As there is no convincing data suggesting the differences in pore properties between in insect cell lines and midgut cells, we consider we do not need to refer that.
There is no mention or explanation of Cry1Ab not following the linear relationship in the text. Also, one would argue that the model also does not work well for Cry1Ac, as the observed toxicity is an order of magnitude lower than the predicted. These observations suggest the model does not accurately predict toxicity for 2 out of 3 C2/Cad-Cry1A toxins tested. Differences of an order of magnitude in an in vitro system may translate to important inconsistencies between predicted and observed toxicity in vivo, throwing more doubt on the validity of the model.
>Our argument is “Toxicities predicted using this formula are strongly correlated with the actual synergism-mediated toxicity observed in in vitro assay (R2 = 0.9039, p < 0.001) (Figure 11), indicating that the Cry-receptor binding affinity explains the synergism-mediated toxicity (lines 556-559). We agree that additional factors may be needed to explain the toxicity in C2-Cad/1Ab, but it is also true that, on the whole, this figure indicates a strong correlation with low P-value. Both actual and predicted toxicities are derived from experimental data containing errors, which may cause the prediction error.
Strongly suggest to changing "actual" to "observed" toxicity.
> ”actual” means actual experimental values when contrasting with “predicted” values. “Predicted versus actual plot” is generally used and we do not figure out the reason why the reviewer strongly suggest the change. In most cases we write the text so that readers can see “actual” means experimental data. We modified some sentences to reinforce it: “actual Cry toxicity toward cultured cells coexpressing ABC transporters and cadherin (lines 17-18)” and “the actual synergism-mediated toxicity observed in in vitro assay (line 557)”.
This comment goes back to the observation that the model is developed using in vitro data with a limited set of Cry receptors (Bmcadherin and BmABCC2) and toxins (Cry1A). Additional data with other toxins or insect receptors is needed to support the idea that the model is useful for in vivo predictions. Optimally, one would estimate synergisms with the formula first and then test in vitro to confirm the predictions. Without these data, I do not think the current paper provides enough information to support how applicable the model is to other toxin/insect systems, especially in vivo.
>We agree with that. Both our molecular and kinetic models must be experimentally tested by ourselves and those who are interested, criticized, and revised to be much more sophisticated, or even refuted by novel much more convincing hypothesis. We consider our contribution in this paper is to provide some ideas and hypotheses which could contribute to better understanding of the essence of Cry mode of action.